# Single-cell RNA sequencing and lineage tracing confirm mesenchyme to epithelial transformation (MET) contributes to repair of the endometrium at menstruation

Phoebe M Kirkwood[1], Douglas A Gibson[1], Isaac Shaw[1], Ross Dobie[1], Olympia Kelepouri[1], Neil C Henderson[1,2], Philippa TK Saunders[1]*

[1]Centre for Inflammation Research, University of Edinburgh, Edinburgh, United Kingdom; [2]MRC Human Genetics Unit, Institute of Genetics and Molecular Medicine, University of Edinburgh, Edinburgh, United Kingdom

*For correspondence:
p.saunders@ed.ac.uk

**Competing interest:** The authors declare that no competing interests exist.

**Abstract** The human endometrium experiences repetitive cycles of tissue wounding characterised by piecemeal shedding of the surface epithelium and rapid restoration of tissue homeostasis. In this study, we used a mouse model of endometrial repair and three transgenic lines of mice to investigate whether epithelial cells that become incorporated into the newly formed luminal epithelium have their origins in one or more of the mesenchymal cell types present in the stromal compartment of the endometrium. Using scRNAseq, we identified a novel population of PDGFRb + mesenchymal stromal cells that developed a unique transcriptomic signature in response to endometrial breakdown/repair. These cells expressed genes usually considered specific to epithelial cells and in silico trajectory analysis suggested they were stromal fibroblasts in transition to becoming epithelial cells. To confirm our hypothesis we used a lineage tracing strategy to compare the fate of stromal fibroblasts (PDGFRa+) and stromal perivascular cells (NG2/CSPG4+). We demonstrated that stromal fibroblasts can undergo a mesenchyme to epithelial transformation and become incorporated into the re-epithelialised luminal surface of the repaired tissue. This study is the first to discover a novel population of wound-responsive, plastic endometrial stromal fibroblasts that contribute to the rapid restoration of an intact luminal epithelium during endometrial repair. These findings form a platform for comparisons both to endometrial pathologies which involve a fibrotic response (Asherman's syndrome, endometriosis) as well as other mucosal tissues which have a variable response to wounding.

## Editor's evaluation

The investigators present an important finding showing the contribution of mesenchyme to epithelial transformation (MET) to the healing of the endometrial luminal epithelium, furthering our understanding of the mechanisms underlying endometrial regeneration, with a potential impact on related pathology.

## Introduction

Efficient wound repair and restoration of tissue homeostasis is essential for reproductive and general health. Repetitive injury, inflammation, and fibrosis resulting in disordered tissue architecture is a

**eLife digest** The human uterus is a formidable organ. From puberty to menopause, it completely sheds off its internal lining every 28 days or so, creating what is in effect a large open wound. Unlike the skin or other parts of the body, however, this tissue can quickly repair itself without scarring. This fascinating process remains poorly understood, partly because human samples and animal models that mimic human menstruation are still lacking. This makes it difficult to grasp how various types of uterine cells get mobilised for healing.

To fill this gap, Kirkwood et al. focused on fibroblasts, a heterogenous cell population which helps to support the epithelial cells lining the inside of the uterus. How these cells responded to the advent of menstruation was examined in female mice genetically manipulated to have human-like periods. A method known as single-cell RNAseq was used to track which genes were active in each of these cells before, one day and two days after period onset. This revealed the existence of a subpopulation of cells which only appeared when wound healing was most needed.

These 'repair-specific' fibroblasts expressed a mixture of genes; those typical of fibroblasts but also some known to be active in the epithelial cells lining the uterus. This suggests that the cells were in the process of changing their identity so they could remake the uterine layer lost during a period. And indeed, labelling these fibroblasts with a fluorescent tag showed that, during healing, they had migrated from within the uterine tissue to become part of its newly restored internal surface. These results represent the first evidence that fibroblasts play a direct role in repairing the uterus during menstruation.

From endometriosis to infertility, the lives of millions of people around the world are impacted by disorders which affect the uterine lining. A better understanding of how the uterus can fix itself month after month could help to find new treatments for these conditions. This knowledge could also be useful for to address abnormal wound healing in the skin and other tissues, as this process often involves fibroblasts.

major cause of morbidity and mortality due to organ failure (*Greenhalgh et al., 2015*). The endometrium, a complex hormone responsive multicellular tissue, exhibits remarkable resilience in its ability to respond to the monthly wound of 'menstruation' without fibrosis and with a response that is both rapid and scar-free (*Garry et al., 2009*; *Ludwig and Spornitz, 1991*). A critical 'trigger' for menstruation in women is the rapid fall in circulating concentrations of progesterone that occurs as a result of the involution of the ovarian corpus luteum in a non-fertile cycle (*Maybin and Critchley, 2011*). Menstruation is considered an inflammatory event with a 'cascade' of changes including an increase in synthesis of pro-inflammatory factors such as prostaglandins, cytokines, and chemokines which is accompanied by infiltration of immune cells of the myeloid lineage (neutrophils, monocyte/ macrophages) (*Evans and Salamonsen, 2012*). Increased production of prostaglandins is associated with arteriole vasoconstriction which results in local hypoxia, stabilisation of HIF1alpha and increased expression of genes such as *VEGF* (*Critchley et al., 2006b*). Whilst we have a good appreciation of the mechanisms that contribute to the tightly regulated breakdown and shedding of a portion of the endometrium at menses gaps remain in our understanding of the processes that contribute to rapid, fibrosis/scar-free restoration of an intact luminal epithelium over the tissue surface. It is also important to separate the mechanisms responsible for menstrual repair, which occurs at a time in the cycle when ovarian-derived steroids are low, from endometrial regeneration/proliferation which occurs subsequently in response to rising levels of follicle-derived oestrogens during the proliferative phase of the cycle (*Maybin and Critchley, 2012*).

Several different mechanisms have already been proposed as contributing to the luminal repair process including, activation of progenitor/stem cells in the stroma and/or epithelium, epithelial cell proliferation/migration and transformation of stromal cells into epithelial cells (mesenchyme to epithelial transition, MET) each of which are discussed briefly below.

The ability of the endometrium to rapidly regenerate during each menstrual cycle has prompted researchers to explore a role for putative stem/progenitor cells residing in either the stroma and/ or the epithelial compartments in endometrial repair (*Gargett et al., 2016*; *Cousins et al., 2021*). The Gargett group have identified populations of endometrial mesenchymal stem cells; eMSC with

clonogenic and broad multilineage potential in both human endometrial tissue and menstrual fluid and characterised them as PDGFRB+/CD146+/ SUSD2+ (*Gargett et al., 2016*). Notably in human endometrial tissue sections these cells had a perivascular location and a putative pericyte identity. A role for progenitor cells residing in the basal glandular epithelium has also been proposed with a recent review highlighting a number of marker proteins employed to establish their identity including SOX9 and SSEA1 (*Cousins et al., 2021*). Another source of putative stem cells that may contribute to endometrial repair processes and/or endometrial pathology is the bone marrow (*Kaitu'u-Lino et al., 2012*) although evidence for the contribution of bone marrow-derived stem cells is somewhat limited (*Deane et al., 2019*; *Aghajanova et al., 2010*).

During the first 2 days of menstruation rapid shedding of the luminal portion of the tissue (functional layer) leaves a denuded (basal) stroma interspersed with glandular 'stumps'. This phenomenon was beautifully documented in the elegant studies of *Ludwig and Spornitz, 1991*. In the 1970's Ferenczy proposed that new epithelial cells arose from proliferation of intact surface epithelium bordering denuded stroma and the basal glands that were exposed when the functional layer was shed (*Ferenczy, 1976b*; *Ferenczy, 1976a*). He noted that the re-epithelialisation of the human endometrium was rapid (~48 hr) and did not appear to involve mitosis of the epithelial cells (migration?). These mechanisms of epithelial repair were later endorsed by *Ludwig and Spornitz, 1991* who also observed fibrin mesh formation on the denuded surface in the early days of menstruation and growth of an epithelial monolayer in spirals which would appear to be consistent with them being associated with the residual stumps of endometrial glands. When Garry and colleagues used a more dynamic approach involving pressure-controlled, continuous flow hysteroscopy they found evidence that loss and repair of the surface occur synchronously at different places within the endometrial cavity a process they called 'piecemeal' (*Garry et al., 2009*). They also reported epithelial cells of the glands underwent apoptosis and were shed with the decidual tissue mass. Apoptosis of glandular epithelial cells in carefully dated human endometrium has also been documented using staining for cleaved caspase 3 and shown to be high in both late secretory and menstrual phases (*Armstrong et al., 2017*). Taken together the data from Garry et al and Armstrong et al challenges the earlier papers citing the glands as the source of new luminal cells.

There is also evidence that cells within the stromal compartment may change their identity to adopt an epithelial phenotype (mesenchyme to epithelial transition, MET) and thereby contribute to the luminal surface. This was first proposed in the 1960's by Baggish et al who evaluated sections of menstruating human endometria and noted hyperchromatic stromal cells between gland stumps that appeared to take part in re-epithelisation by a process simulating metaplasia (*Baggish et al., 1967*). In the 1970's the involvement of the stroma was challenged (*Ferenczy, 1976b*) but has been supported by more recent studies (*Garry et al., 2009*). Specifically in 2009 the authors reported that the denuded basalis endometrium was rapidly covered with a fibrinous mesh within which new surface epithelial cells were present which appeared to arise from the underlying stroma.

To complement studies on human tissue samples and studies in primates (*Brenner et al., 2002*; *Critchley et al., 2006a*) we have used a highly reproducible mouse model that recapitulates key features of human menstruation in response to withdrawal of progesterone including vascular breakdown (vaginal bleeding), spatial and temporal hypoxia, cell apoptosis, expression of matrix metalloproteinases and influx of myeloid immune cells (*Armstrong et al., 2017*; *Maybin et al., 2018*; *Cousins et al., 2016a*; *Cousins et al., 2016b*; *Cousins et al., 2014*). Importantly, as in the human endometrium (*Garry et al., 2009*), restoration of an intact luminal epithelial layer occurs within two days (*Cousins et al., 2014*). Consistent with some of the reports on repair processes in human endometrium we found evidence for epithelial cell migration and proliferation of residual epithelial cells (Ki67+) but no significant proliferation of basal glands (*Cousins et al., 2014*). Other groups have also used similar mouse models to explore the dynamic changes during a menstrual-like event. Kaitu'u-Lino et al used a mouse model in combination with pulse labelling of cells with BrdU, and reported that a population of epithelial progenitor cells residing in the basal glands might contribute to postmenstrual repair (*Kaitu'u-Lino et al., 2010*, *Kaitu'u-Lino et al., 2012*).

In our endometrial repair/menstrual model one of our novel findings was the existence of stromal cells that co-expressed vimentin (stromal cell marker) and cytokeratin (epithelial cell marker) specifically residing in areas denuded of epithelium which suggested to us that stromal cells might be changing their phenotype via MET (*Cousins et al., 2014*).

In the current study, we have used single cell RNA sequencing and lineage tracing to specifically address which, if any, of the stromal mesenchymal cell subpopulations we previously identified in the mouse endometrium (*Kirkwood et al., 2021*) might contribute to the rapid restoration of the intact luminal epithelial cell layer observed in our mouse model (*Cousins et al., 2014*; *Cousins et al., 2016a*). To the best of our knowledge this is the first time that inducible-Cre lineage tracing approaches in adult mice have been used to explore MET in a model of menstruation.

## Results

### Single-cell RNA sequencing identifies novel mesenchymal cell subpopulations unique to 'repairing' tissue in a mouse model of endometrial breakdown and repair (menstruation)

To identify whether changes in the transcriptome of mesenchymal cells occurred during endometrial repair we recovered tissues from mice expressing the *Pdgfrb*-BAC-eGFP transgene during the normal cycle (controls; n=4) and at 24 (n=4) and 48 hr (n=4) after progesterone withdrawal using a well-validated model of endometrial shedding/repair (simulated 'menstruation'; *Cousins et al., 2016b*; *Figure 1A* i, ii). Population restricted cell sorting was used to isolate CD31-/CD45-/GFP+(PDGFRβ+) cells prior to single-cell sequencing. In cycling mice we have previously reported GFP + cells are restricted to the stromal compartment of the endometrium (*Kirkwood et al., 2021*).

scRNAseq was used to generate a dataset for each timepoint and these were then integrated for downstream cluster and differential gene expression analysis. This strategy resulted in identification of 8 transcriptionally distinct clusters of GFP + cells (*Figure 1B*). The expression of canonical phenotypic gene markers for mesenchymal, perivascular and fibroblast cell populations previously identified in control/cycling mouse endometrium was used to assign putative identities to each cluster (*Kirkwood et al., 2021*). We identified three perivascular populations (V, P1, P2; expression of *Mcam*, *Cspg4*, *Rgs4*, *Acta2*) and five fibroblast subpopulations (F1-5; expression of *Pdgfra*, *Cd34*, *Fbln1/2*, *Mfap4/5*, *Col1a1*). Importantly, cells in all eight clusters expressed mesenchymal markers *Pdgfrb*, *Vim*, *Des*, and *Thy1* (*Figure 1C*, *Figure 2—figure supplement 1A(iii)*) while no clusters expressed *Pecam1* (CD31) or *Ptprc* (CD45) (*Figure 2—figure supplement 1A(i-ii)*) consistent with the cell isolation strategy employed.

Analysis of the distribution of these subpopulations in the different datasets (*Figure 1D*) confirmed the presence of five mesenchyme subpopulations previously identified in cycling mouse uterine tissue (V, P1, F1-3; *Kirkwood et al., 2021*). However in tissue recovered at 24 and 48 hr three additional transcriptionally distinct repair-specific cell clusters were identified: P2 and F4 in 24 hr tissues (*Figure 1D*; green and purple circles respectively) and F5 present in 48 hr tissues (*Figure 1D*; pink ellipse).

### Gene expression analysis identified a unique repair-associated cluster that expressed genes associated with epithelial cell phenotype

Differential gene expression analysis revealed a very high degree of similarity between P1 (previously identified as pericytes) (*Kirkwood et al., 2021*) and P2 which was specific to 24 hr tissue (*Figure 2A*). Unbiased GO analysis of the P2 transcriptome identified functions associated with blood vessel formation and immune cell signalling, characteristic of pericytes (*Figure 2—figure supplement 1D(i)*) suggesting a role for these cells in angiogenesis and regulation of the immune system in response to changes in the local environment.

We noted that cluster F5 (specific to 48 hr tissue) expressed a large number of mitochondrial genes and genes associated with cell death including *Casp3*, *Casp9*, *Trp53* and *Bax* (*Figure 2—figure supplement 1*). GO analysis of the F5 transcriptome identified functions associated with DNA damage. Based on this in silico data and the results in our previous study using the same mouse model system that detected high levels of cleaved caspase expression in the stromal compartment at 24 hr (*Armstrong et al., 2017*) we suggest that F5 represents a cluster of dead/apoptotic cells and it was therefore excluded from further downstream analyses.

Cluster F4 (24 hr tissue) was of particular interest as it had a transcriptomic signature that was distinct from to those of the mesenchymal cell populations (F1-3) present in cycling endometrium (*Kirkwood et al., 2021*; *Figure 2A*; red box). Unbiased GO analysis enriched for biological functions that are associated with endometrial tissue repair including response to wounding, reproductive

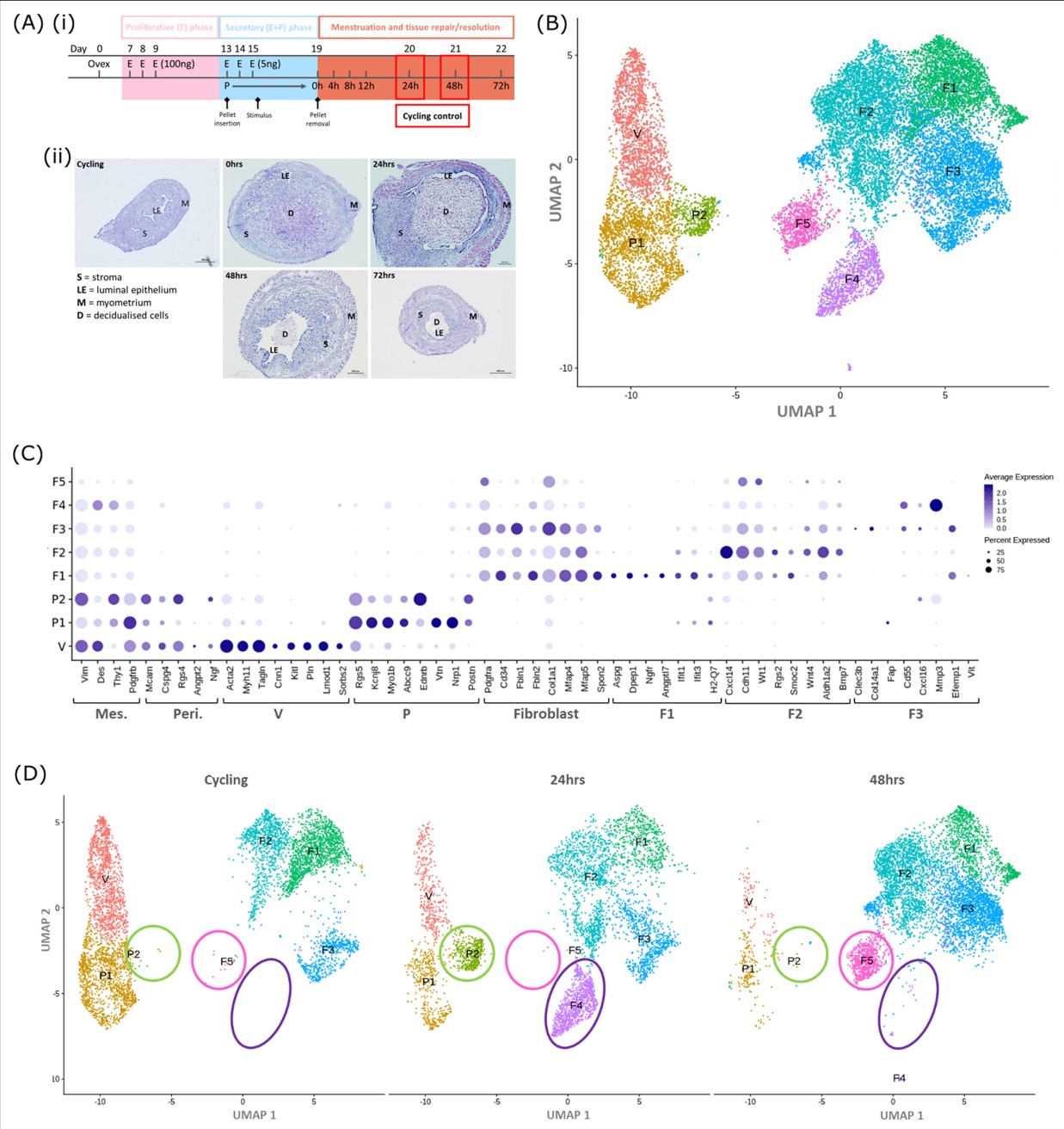

**Figure 1.** Single-cell RNA sequence analysis identified mesenchymal cell populations unique to 'repairing' tissue in a mouse model of endometrial breakdown and repair (menstruation). (**A, i**) Mouse model of endometrial tissue breakdown and repair, (ii) histological morphology of uterine tissues 0, 24, 48, and 72 hr progesterone withdrawal illustrating tissue breakdown, repair, remodelling and resolution. (**B**) UMAP visualisation: GFP +mesenchymal cells isolated from *Pdgfrb*-BAC-eGFP mouse endometrium (cycling plus 24/48 hr after progesterone withdrawal) cluster into eight distinct populations. (**C**) Dot plot: expression of canonical gene signatures associated with known cell types present in the endometrium: mesenchymal cells (*Pdgfrb*, *Vim*, *Des*, *Thy1*), perivascular cells (*Mcam*, *Cspg4*, *Rgs4*, *Acta2*) and fibroblasts (*Pdgfra*, *Cd34*, *Fbln1/2*, *Mfap4/5*, *Col1a1*) (dot colour: average expression per cluster; dot size: percent cluster expressing gene) (**D**) UMAP visualisation: mesenchymal clusters split by source/dataset identifies transient repair-specific subpopulations P2 and F4 in 24 hr tissues and F5 in 48 hr tissues.

structure development, response to oxidative stress and regulation of vasculature development. Additional functions that were identified included epithelium migration, morphogenesis of a branching epithelium and regulation of epithelial cell migration (*Figure 2B*).

Cluster F4 was the only mesenchymal cell cluster that expressed markers known to be associated with both definitive mesenchymal (*Pdgfrb*, *Vim*) and epithelial (*Epcam*, *Krt18*) cell types (*Figure 2C*).

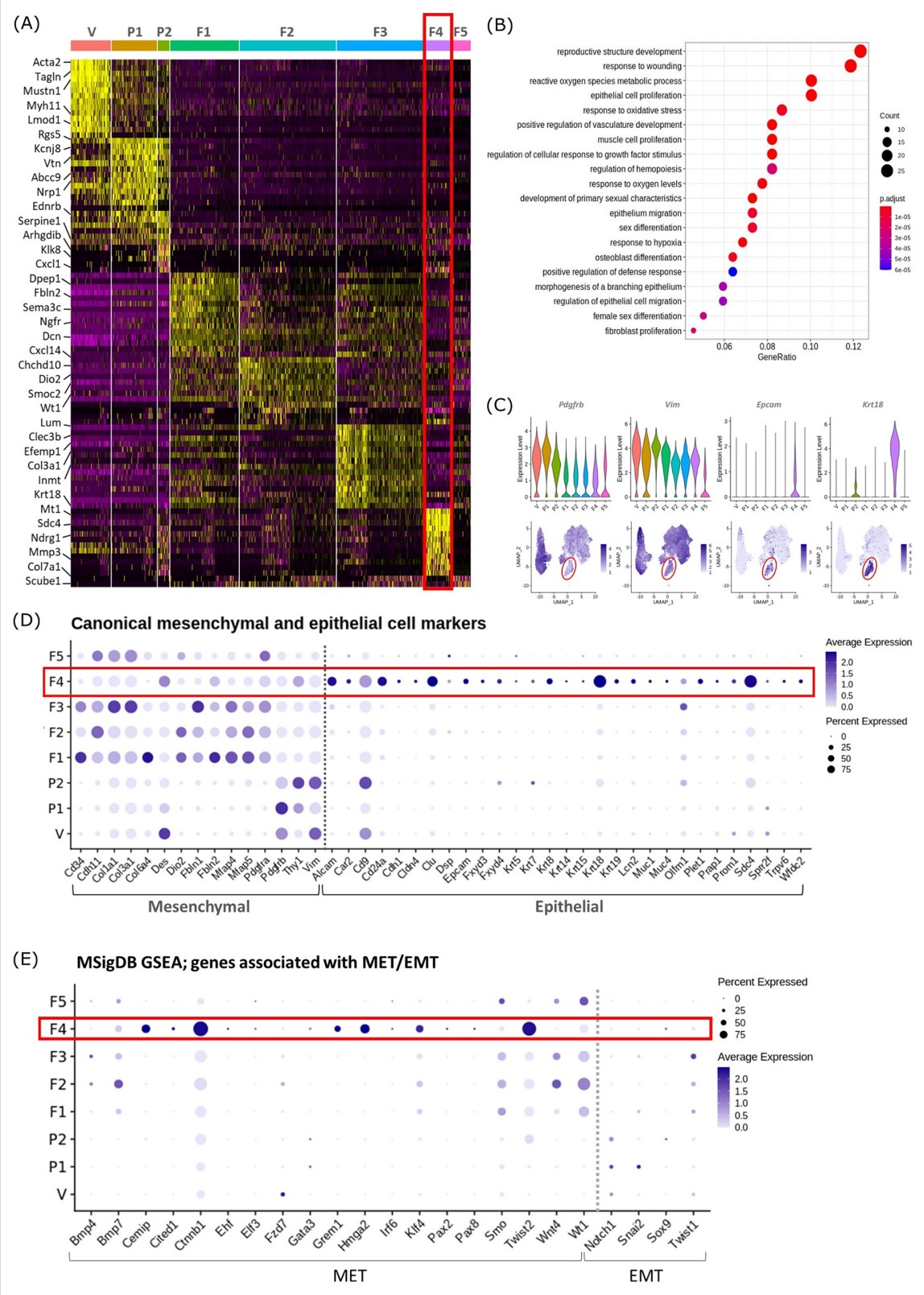

**Figure 2.** Gene expression analysis identifies transcriptomic profiles of mesenchymal cells and the expression of genes associated with epithelial cell identity in repair-specific fibroblasts (cluster F4). (**A**) Heatmap (yellow, high; purple, low) displaying differentially expressed genes per cluster when compared to all other clusters (logFC >0.5, pvalue <0.05, Wilcoxon rank-sum test) top is colour coded and named by cluster; V=vascular smooth muscle cells (vSMCs), P1=pericytes 1, P2=pericytes 2, F1=fibroblasts 1, F2=fibroblasts 2, F3=fibroblasts 3, F4=fibroblasts 4, F5=fibroblasts 5. The expression

*Figure 2 continued on next page*

*Figure 2 continued*

of the top 5 exemplar genes in each cell cluster is illustrated. (**B**) Dot plot: GO enrichment terms relating to biological processes (BP) associated with the genetic signature of repair-specific cluster F4 (dot size: gene ratio, number of genes in data/number of genes associated with GO term; dot colour: p-value representing the enrichment score). (**C**) Gene expression plots: expression of canonical mesenchymal cell markers (**i**) *Pdgfrb* and (**ii**) *Vim* and canonical epithelial cell markers (**iii**) *Epcam* and (**iv**) *Krt18*. Note F4 fibroblasts (red box) express markers associated with both cell lineages. (**D**) Dot plot: expression of extended gene signatures associated with mesenchymal and epithelial cell lineages in the endometrium (dot colour: average expression per cluster; dot size: percent cluster expressing gene). (**E**) Dot plot: expression of genes associated with the regulation of MET/EMT taken from the MSigDB (dot colour: average expression per cluster; dot size: percent cluster expressing gene).

The online version of this article includes the following figure supplement(s) for figure 2:

**Figure supplement 1.** Gene expression data for mesenchyme subpopulations including identification of apoptosis related genes.

---

Comparison across all mesenchymal clusters of the expression of both known canonical mesenchymal/fibroblast markers including *Thy1, Cd34, Cdh11, Col1a1, Col3a1, Fbln1, Fbln2, Mfap4,* and *Mfap5* and epithelial markers such *as Krt8, Krt19, Muc1, Muc4, Cd24a, Car2, Sdc4, Alcam,* and *Cdh1* revealed F4 to be the only cluster that expressed genes associated with both cell types. Notably, F4 did not express the full array of keratins and mucins associated with definitive epithelial lineages but preferentially expressed Alcam, Cd24a, Clu, Krt18, and Sdc4 when compared to other mesenchymal clusters (*Figure 2D*; red box). We considered this was evidence that these cells may be of an intermediate phenotype and represent cells undergoing a mesenchymal to epithelial transition (MET) but not yet fully committed to a definitive epithelail lineage. To investigate this further we accessed annotated gene sets from the MSigDB (*Subramanian et al., 2005*) that are known to be associated with the regulation of MET/EMT. This analysis identified that F4 also expressed key transcription factors associated with MET/EMT including *Cemip, Cited1, Ctnnb1, Grem1, Hmga2, Klf4,* and *Twist2* (*Figure 2E*).

## Cells that co-express both mesenchymal and epithelial cell markers can be detected in the uterus during endometrial tissue repair

To validate this finding and to analyse the topography of putative MET cells in the mouse endometrium, sections from *Pdgfrb*-BAC-eGFP mice were co-stained with antibody directed against EPCAM. Cells that expressed both GFP and EPCAM were identified in endometrial tissue undergoing active repair (24 hr; *Figure 3A* (i); white arrows) specifically in regions where (a) the stromal surface was denuded after decidual shedding and (b) a new epithelial cell layer was being re-instated. Some GFP + EPCAM + cells persisted in the luminal layer at 48 hr but were not detected in 72 hr tissue (*Figure 3A* ii, iii). Consistent with previous data in control/cycling endometrium expression of the GFP reporter was exclusive to the stromal compartment and epithelial cells were EPCAM + (yellow arrows, *Figure 3A* iv; *Kirkwood et al., 2021*).

Analysis of the same tissues using flow cytometry confirmed the presence of a new population of cells that expressed both GFP and EPCAM at 24 hr with a reduction in their number at 48 hr and levels that appeared comparable to cycling/control tissues in the fully remodelled tissue (72 hr) (*Figure 3B–C*). These data were in agreement with the results of scRNAseq analysis where *Epcam* was detected in the F4 cluster in the 24 hr dataset (*Figure 2C*) with analysis of the tissue sections suggesting the F4 cluster may represent a mixture of cells in the stroma in the process of undergoing MET (*Figure 3Aa*) as well as those which are already incorporated into the luminal epithelium (*Figure 3Aa*) where they are still detectable at 48 hr (*Figure 3Aii*). Whilst promising studies on cell fate using mice from the *Pdgfrb*-BAC-eGFP 'knock-in' line has limitations as expression of GFP is limited to cells in which the *Pdgfrb* is active and immunohistochemistry has revealed that the PDGFRB protein is only present in endometrial mesenchyme (see Figure 1 in *Kirkwood et al., 2021*). With high expression of *Pdgfrb* in the mesenchyme and some expression persisting in F4 (*Figure 2C*) we speculated that presence of GFP in the luminal epithelial cells at 24 and 48 hr represented mesenchyme cells that underwent a rapid MET. It was not possible to determine whether these cells did/or did not persist at 72 hr as absence of GFP might be due to protein turnover and therefore additional studies with new mouse lines were necessary to resolve this question.

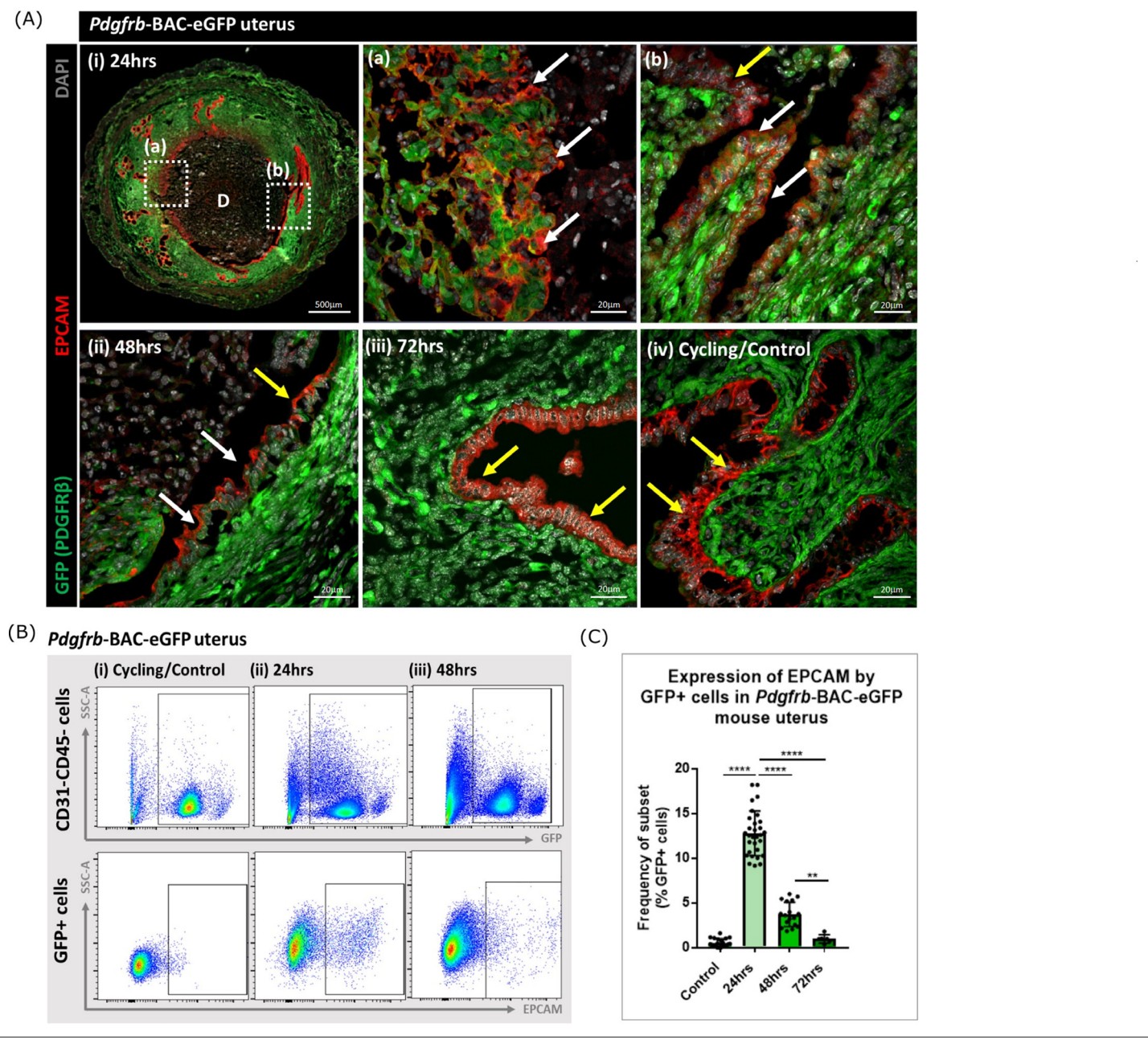

**Figure 3.** Analysis of *Pdgfrb*-BAC-eGFP uterine tissue during endometrial tissue repair and remodelling identifies transient expression of GFP in EPCAM + epithelial cells. (**A**) Immunohistochemical analysis of GFP reporter protein (green) and epithelial cell marker EPCAM (red) in uterine tissues 24, 48, and 72 hr following progesterone withdrawal. (**i**) In 24 hr tissues cells that are GFP + EPCAM + are detected in (**a**) regions of denuded stromal surfaces where the decidualised tissue has detached (white arrows) and (**b**) regions where a new epithelium has formed (white arrows). (ii) At 48 hr GFP + EPCAM + cells were detected within the renewed luminal epithelium (white arrows) adjacent to GFP- EPCAM + epithelial cells (yellow arrows). (iii) At 72 hr no GFP + EPCAM + cells were detected. (iv) In control/cycling tissues epithelial cells are EPCAM + (yellow arrows), stroma is GFP+. (**B**) Flow cytometry (FC): analysis of EPCAM in *Pdgfrb*-BAC-eGFP uterine tissues at 24, 48, and 72 hr following progesterone withdrawal; note detection of a new population of GFP + EPCAM + cells at 24 hr (boxed lower panel) with decreased numbers at 48 hr. (**C**) Bar plot: quantification of FC data analysing the expression of EPCAM by GFP reporter positive cells in the uterus (%) - control (n=21) 0.53 ± 0.46%; 24 hr (n=30) 12.78 ± 2.49%; 48 hr (n=17) 3.71 ± 1.36%; 72 hr (n=6) 0.96 ± 0.47%; one-way ANOVA, Sidak's multiple comparisons.

The online version of this article includes the following source data for figure 3:

**Source data 1.** Summary statistics for flow cytometry analyses performed in *Figure 3C*.

**Source data 2.** One-way ANOVA with Sidak's multiple comparisons test for *Figure 3C*.

## Comparison between fibroblast and epithelial cell scRNA clusters confirmed the repair-specific cells (F4) had a unique transcriptome

To compare the transcriptome of the F4 cluster (putative MET mesenchyme/epithelium) to differentiated endometrium epithelial cells scRNAseq datasets were generated by from CD31-/CD45-/GFP-/EPCAM + epithelial cells obtained from the *Pdgfrb*-BAC-eGFP mouse uterus (control/cycling and 48 hr). Bioinformatic analysis combining GFP + datasets (cycling/24 hr/48 hr) and GFP-EPCAM + epithelial cell data (cycling/48 hr) identified 16 distinct cell clusters (***Figure 4A and C***). Based on expression of *Pdgfrb* (***Figure 4Bii***) and canonical markers (***Figure 4D***) seven of these had a mesenchyme phenotype: three perivascular (V, P1-2), four fibroblast (F1-4). Nine of the clusters expressed classical markers of epithelial cells including *Epcam* (E1-9) (***Figure 4A and B*** (i, ii), C). Notably, the previously identified repair-specific fibroblasts (F4) formed a cluster which appeared in close proximity to both fibroblast and epithelial cell clusters but remained distinct from definitive perivascular cells (V, P1, P2). Doublet analysis was also conducted to confirm this was a unique population (***Figure 4—figure supplement 1A***).

A high degree of similarity was observed between the transcriptomic signature of F4 and the definitive epithelial cell clusters E1 and E2 (***Figure 4C***, red box, ***Figure 4—figure supplement 1B***). Extended analysis of canonical markers of mesenchymal (perivascular/fibroblast) and epithelial cell types confirmed that cluster F4 uniquely expressed genes associated with both fibroblast and epithelial lineages (***Figure 4D***, ***Figure 4—figure supplement 1C(i-iv)***). Additional analysis of genes associated with different epithelial subtypes present in the endometrium inferred putative epithelial phenotypes to each of the clusters: E1/2/3 luminal/differentiating, E4 proliferative, E5/6 glandular/ciliated, E7 secretory, E8/9 basal (***Figure 4E***). Notably, the repair-specific mesenchyme cluster F4 also expressed genes associated with both luminal (*Fxyd4, Gsto1, Plac8, Ctsl, Lgals3, Cited2*) and differentiating (*Ctnnb1, Krt8, Krt18, Plk2, Basp1, Ptgs2, Olfm1*) epithelial cell phenotypes with no expression of genes considered definitive for glandular epithelium (*Foxa2, Cxcl15*).

## In silico trajectory analysis infers a putative differentiation trajectory between fibroblasts and epithelial cells

Based on the pattern of gene expression and clustering of scRNA datasets generated from the purified mesenchyme and epithelial cells populations, we made a preliminary assessment that the F4 population were an intermediate cell type. To investigate this further, we used the Monocle3 package to perform in silico lineage tracing identify whether an inferred differentiation trajectory existed between the different cell clusters.

Monocle3 analysis segregated the scRNAseq data into two partitions: one that encompassed the perivascular subsets; and one that contained both the fibroblast and epithelial subsets. We infer that this would be consistent with a pattern of genetic changes linking fibroblasts and epithelial cells but none between either fibroblasts or epithelial cells and the perivascular clusters (***Figure 5A*** i-ii). Further interrogation of the Monocle data identified a 'root' within fibroblast cluster F2 with a branch moving through the repair-specific cluster F4 into a root in the epithelial clusters on the border of E1 and E2, with subsequent branches moving into other epithelial subsets (***Figure 5A*** iii). No roots were placed within the F4 cluster (only a trajectory branch; ***Figure 5A*** i-iii) which further suggests this population does not have a terminally differentiated phenotype, rather it represents a transitional transcriptional state. We used the *scVelo* R package to interrogate this putative trajectory by calculating cellular velocity from spliced and unspliced mRNA content. A transition from F2 to F4 was observed, inferring the presence of pseudotemporal dynamics between the two subpopulations, with cells in the F4 cluster displaying a positive unspliced/spliced ratio reinforcing their potential for cellular transition in the direction away from F2 and towards the epithelial clusters (E1/2) (***Figure 5—figure supplement 1A, B***; red asterisk). Similarly, a transition from P1 to P2 was observed (***Figure 5—figure supplement 1A, B***; blue asterisk). Notably, the transitions within epithelial sub-populations E1/2 appeared to move away from F4 and towards E1 and E2 (***Figure 5—figure supplement 1A, B***; green asterisk). Taken together, these bioinformatic analyses suggest that a cellular transition from F1/2/3 through F4 into E1/2 may exist.

The results described above would be consistent with the presence of a new population of fibroblast origin that has a distinct transcriptional profile that exists as a transient/intermediate population in response to the altered environment that was induced by progesterone withdrawal. To confirm the

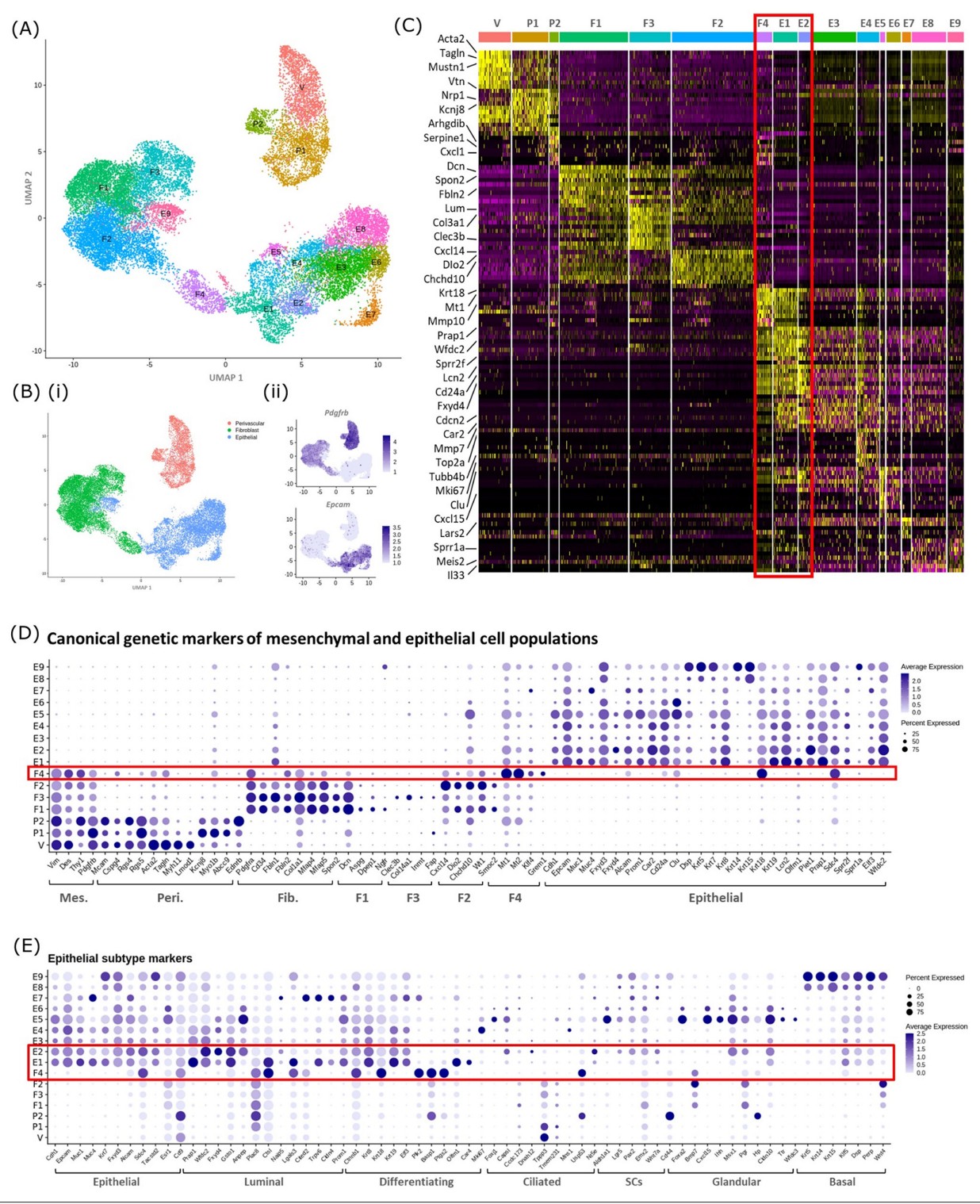

**Figure 4.** In silico analysis of scRNAseq reveals transcriptomic similarity between repair-specific mesenchymal cells and subpopulations of EPCAM + endometrial epithelial cells. (**A**) UMAP visualisation: GFP + mesenchymal cells and GFP-EPCAM + epithelial cells isolated from *Pdgfrb*-BAC-eGFP mouse endometrium (cycling/24/48 hr and cycling/48 hr, respectively) cluster into 16 distinct populations: 7 mesenchymal, 9 epithelial. Repair-specific fibroblasts (**F4**) cluster in close proximity to epithelial cell clusters but remain a distinct population of cells. (**B**) (**i**) UMAP visualisation: three groups of cell populations are detected in the data: perivascular, fibroblast, and epithelial cells (add colours) (**ii**) gene expression of *Pdgfrb* is restricted to mesenchymal populations (perivascular/fibroblast) while *Epcam* is restricted to epithelial populations (and repair-specific fibroblasts; **F4**), confirming the isolation strategy adopted. (**C**) Scaled heatmap (yellow, high; purple, low) displaying differentially expressed genes per cluster when compared to all

*Figure 4 continued on next page*

*Figure 4 continued*

other clusters (logFC >0.5, pvalue <0.05, Wilcoxon rank-sum test) top is colour coded and named by cluster; V=vascular smooth muscle cells (vSMCs), P1/2=pericytes 1/2, F1−4=fibroblasts 1−4, E1−9=epithelial cells 1−9. The expression of 3 exemplar genes in each cell cluster is displayed. F4 shows genetic similarity to E1/2 (red box). (**D**) Dot plot: expression of canonical genes associated with mesenchymal, perivascular, fibroblast and epithelial lineages, F4 is the only cluster that expressed genes from multiple lineages (red box) (dot colour: average expression per cluster; dot size: percent cluster expressing gene). (**E**) Dot plot: expression of gene signatures associated with known epithelial cell subtypes present in the endometrium as per the literature: canonical epithelial, luminal, differentiating, ciliated, SCs, glandular, basal epithelial cells (dot colour: average expression per cluster; dot size: percent cluster expressing gene). Red box F4 compared to E1, E2 highlighting expression of genes found in the luminal and differentiating epithelial cell populations.

The online version of this article includes the following figure supplement(s) for figure 4:

**Figure supplement 1.** Comparison between gene expression in F4 population and all other cell clustes confirms overlap with both fibroblast and epithelial cells.

origin and fate of these cells we adopted a new strategy using in vivo genetic fate-mapping based on induction of cre recombinase under the control of promoters specific to genes we have identified as specific for fibroblast (*Pdgfra*) and perivascular (*Cspg4*/NG2) cell clusters (*Figure 5B and C*; *Kirkwood et al., 2021*).

## Induction of tdTm transgenes using an iCRE strategy to separately target fibroblast and perivascular cell populations

To validate the in silico trajectory analysis, we used an in vivo genetic fate-mapping strategy using the TMX-inducible cre recombinase system (*Figure 5B*). Based on our scRNAseq, all mesenchymal cell populations express *Pdgfrb* but we were able to discriminate between fibroblasts (F1-F4) and perivascular cells based on expression of *Pdgfra* and *Cspg4*/NG2 respectively (*Figure 5C*). Importantly neither of these markers were expressed in any of the epithelial cell clusters. We specifically targeted these two mesenchymal cell types in adult female mice using a tamoxifen induction strategy (*Pdgfra-creERT2/Rosa26-tdTm;* iNG2-creER/*Rosa26-tdTm* mice respectively).

Prior to lineage-tracing studies conducted experiments to optimise the amount of TMX administered both to confirm long term expression of the tdTM protein transgenes but also that the endometrial tissue of the TMX-treated mice was still able to respond appropriately to E2 and the decidualization stimulus (*Figure 5—figure supplement 2*). Specifically, we compared the uterine response (weight, stromal-epithelial cell ratio, decidualization respons following 3 x daily oral gavage with 20 mg/ml TMX) 24 h, 1 w, 4 w and 8 w after the last dose; controls had no TMX. In line with expectations treatment with TMX, which is known to act as an oestrogen receptor agonist in endometrial tissue (*Gielen et al., 2008*), increased uterine weights at 24 hr, this increase was still noticeable at 1 w but had returned to baseline by 4 w (*Figure 5—figure supplement 2*). The weight change was associated with an increase in EPCAM + epithelial cells which also returned to baseline by 4 w. Further tests showed decidualization response and changes in weights following progesterone withdrawal were the same as controls if mice had been treated with TMX 4 w previously and no expression of tdTm was detected in Cre-negative TMX treated mice (*Figure 5—figure supplement 2G-J*).

Experiments to confirm the identity of the stromal cells in both iCre models and to investigate their fate following induction of endometrial breakdown/repair were all conducted using 3 x daily gavage with TMX followed by a 4 w washout period (*Figure 5G*). In the Pdgfra-creERT2/Rosa26-tdTm mouse uterus tdTm expression was detected in cells dispersed throughout the endometrial stromal compartment, but not in cells of the myometrium or luminal/glandular epithelium (*Figure 5D* ii). This pattern replicated the location of GFP^dim stromal fibroblasts in the *Pdgfrb*-BAC-eGFP mouse uterus (*Figure 5D* i) We confirmed co-localisation of tdTm and native protein PDGFRα (*Figure 5E*). We also confirmed that these tdTm + cells did not express CD146 (*Figure 5—figure supplement 2*) and concluded that the tdTM transgene is specific to PDGFRα+endometrial fibroblasts.

In the iNG2-creER/*Rosa26-tdTm* mouse uterus tdTm was detected in groups of cells dispersed throughout the stromal compartment but not the myometrium or epithelium (luminal/ glandular) (*Figure 5D* iii). This pattern was similar to perivascular GFP^bright cells in the *Pdgfrb*-BAC-eGFP mouse uterus (*Figure 5D* (i); *Kirkwood et al., 2021*), and therefore appeared to represent perivascular cells. To confirm this we co-localised the tdTm reporter with native NG2 protein (*Figure 5F*). We also confirmed they express the canonical pericyte/vSMC marker CD146 but not the endothelial cell marker

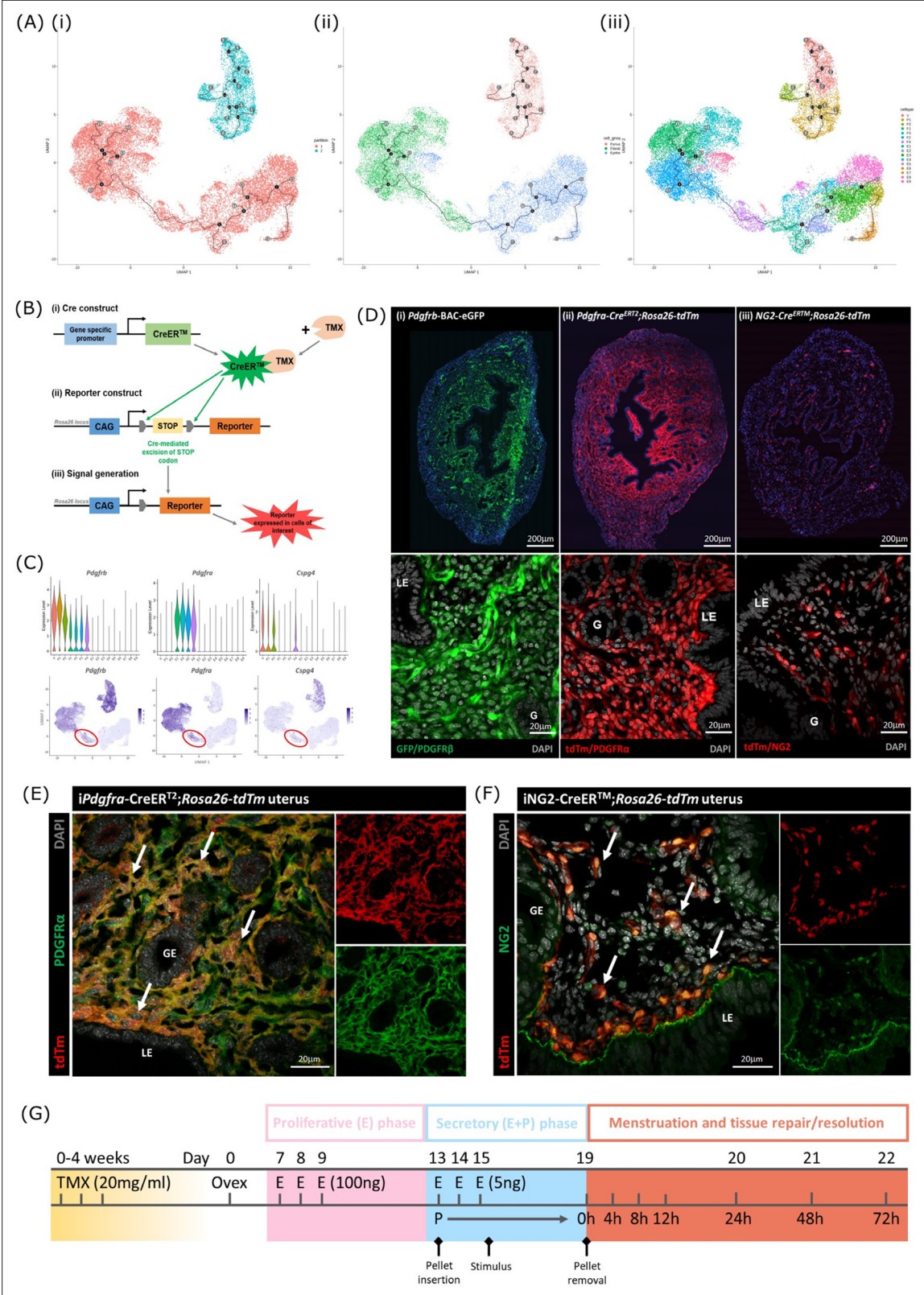

**Figure 5.** Lineage tracing strategy using inducible cre recombinase system to target mesenchymal subpopulations in endometrium of adult mice. (**A**) UMAP visualisation of trajectory analysis. (**i**) Monocle3 separated data into two 'partitions' placing fibroblasts and epithelial cells into the same trajectory. (ii) Monocle3 revealed a putative differentiation trajectory between fibroblasts (green) and epithelial cells (blue) but with no link to perivascular cells (pink). (iii) The trajectory between fibroblasts and epithelial cells runs through the repair-specific fibroblast cluster F4 (black dots: trajectory 'roots'; black

*Figure 5 continued on next page*

*Figure 5 continued*

lines: trajectory 'branches'; grey dots: trajectory 'end-points'). (**B**) Schematic: representation of induced reporter expression in cells of interest using the tamoxifen inducible cre recombinase system. (**C**) Gene expression plots (scRNAseq data): (**i**) *Pdgfrb* all mesenchymal cell clusters (ii) *Pdgfra* fibroblasts, (iii) *Cspg4*/NG2 perivascular cells no expression of these genes in epithelial cell populations. (**D**) Expression of reporter proteins: (**i**) GFP + mesenchyme in *Pdgfrb*-BAC-eGFP mouse endometrium (see *Kirkwood et al., 2021*); (ii) tdTm detected in fibroblasts in *Pdgfra*-creERT2;*Rosa26*-tdTm mouse endometrium; (iii) tdTm detected in perivascular cells in iNG2-Cre^ERTM;*Rosa26*-tdTm mouse endometrium. Epithelial cells (LE/G) do not express reporter proteins any of these mice (LE = luminal epithelium, G=epithelial glands). (**E**) Expression of tdTm reporter and native PDGFRα protein in *Pdgfra*-creERT2;*Rosa26*-tdTm mouse endometrium, co-localisation of tdTm and PDGFRα detected throughout (white arrows). (**F**) Expression of tdTm reporter and native NG2 protein in iNG2-Cre^ERTM;*Rosa26*-tdTm mouse endometrium, co-localisation of tdTm and NG2 detected throughout (white arrows). (**G**) Schematic: mouse model of endometrial tissue breakdown and repair modified to include a 3 x injections with TMX followed by a 4 w washout period prior to ovariectomy.

The online version of this article includes the following figure supplement(s) for figure 5:

**Figure supplement 1.** Trajectory Analysis.

**Figure supplement 2.** Optimisation of reporter gene expression and wash out following induction of transgenes with Tamoxifen.

CD31. Taken together, we concluded that induced tdTm expression in the iNG2-creER/*Rosa26-tdTm* mouse uterus correctly targets NG2 + perivascular cells (pericytes/vSMC).

## Lineage tracing in adult mice confirms MET by PDGFRα+ fibroblasts during endometrial tissue repair

Following transgene induction in *Pdgfra*-creERT2/*Rosa26*-tdTm mice tdTm was detected in cells throughout the stromal compartment in both controls (*Figure 5D*) and at 24 hr after progesterone withdrawal (*Figure 6A*). Notably in the 24 hr tissue a unique population of tdTm + cells that co-expressed the epithelial cell marker EPCAM (tdTm + EPCAM + cells) were detected in two regions one where the decidualised tissue had detached leaving the stromal surface denuded (*Figure 6A* i-iii; white arrows) and the second population having a phenotype that resembled a new epithelial layer which in some cases was continuous with tdTm-EPCAM + cells (*Figure 6A* iv; white and yellow arrows respectively). In other regions epithelial cells that did not express tdTM were present (*Figure 6A* v; yellow arrows).

For independent validation of the tdTm staining of tissue sections flow cytometry was used to quantify the numbers of tdTm + cells that expressed the mesenchymal marker CD90 and/or the epithelial marker EPCAM in controls (no repair) and in the repair model at 24, 48, and 72 hr (*Figure 6B*; red boxes). In controls tdTm + cells were all CD90+ (*Figure 6Bi*) A transient population of tdTm + CD90+EPCAM + cells were detected at 24 hr (red box) which represented ~20% of the reporter + population (*Figure 6C*). Notably whilst some of the tdTM + cells continued to co-express EPCAM at 48 or 72 hr tissue these cells were CD90- (*Figure 6C* iii-iv).

When identical analysis methods were applied to tissue recovered from the NG2-creER/*Rosa26*-tdTm mice cells with a tdTm + CD90+phenotype were identified in controls and at 24, 48, and 72 hr after induction of repair (*Figure 6D*) however using sensitive flow cytometry no tdTm + EPCAM + cells were detected nor were any tdTm + cells present in the epithelial cells at any timepoint (*Figure 6D–F*).

## Lineage tracing confirms that epithelial cells originating from PDGFRα+ fibroblasts are retained within the repaired luminal epithelium

To determine whether stromal fibroblasts we had detected undergoing MET in the *Pdgfra*-creERT2/*Rosa26*-tdTm persisted in the luminal epithelium once it was fully restored (48–72 hr) we conducted a detailed analysis of tissue sections and additional flow cytometry. Notably we identified tdTm + cells in the luminal epithelium at 48 hr adjacent to tdTm- epithelial cells (*Figure 7A* (i-iv); white and yellow arrows respectively). These two populations of cells were also detected in uterine tissues at 72 hr (*Figure 7B* iv). Importantly, tdTm expression was not identified in epithelial cell populations at the time of progesterone withdrawal (0 hr; *Figure 7B* (i); yellow arrows). Quantification of tdTm reporter protein in EPCAM +epithelial cells revealed time dependent changes with proportions rising from ~6% at 24 hr to ~17% at 48 hr and ~15% at 72 hr (*Figure 7C*).

The presence of groups of tdTm + EPCAM + adjacent to tdTm-EPCAM + cells in the newly formed luminal epithelium at 48 hr and their persistence at 72 hr is consistent with MET by PDGFRα+fibroblasts and retention/proliferation of existing epithelial cells both contributing to rapid repair.

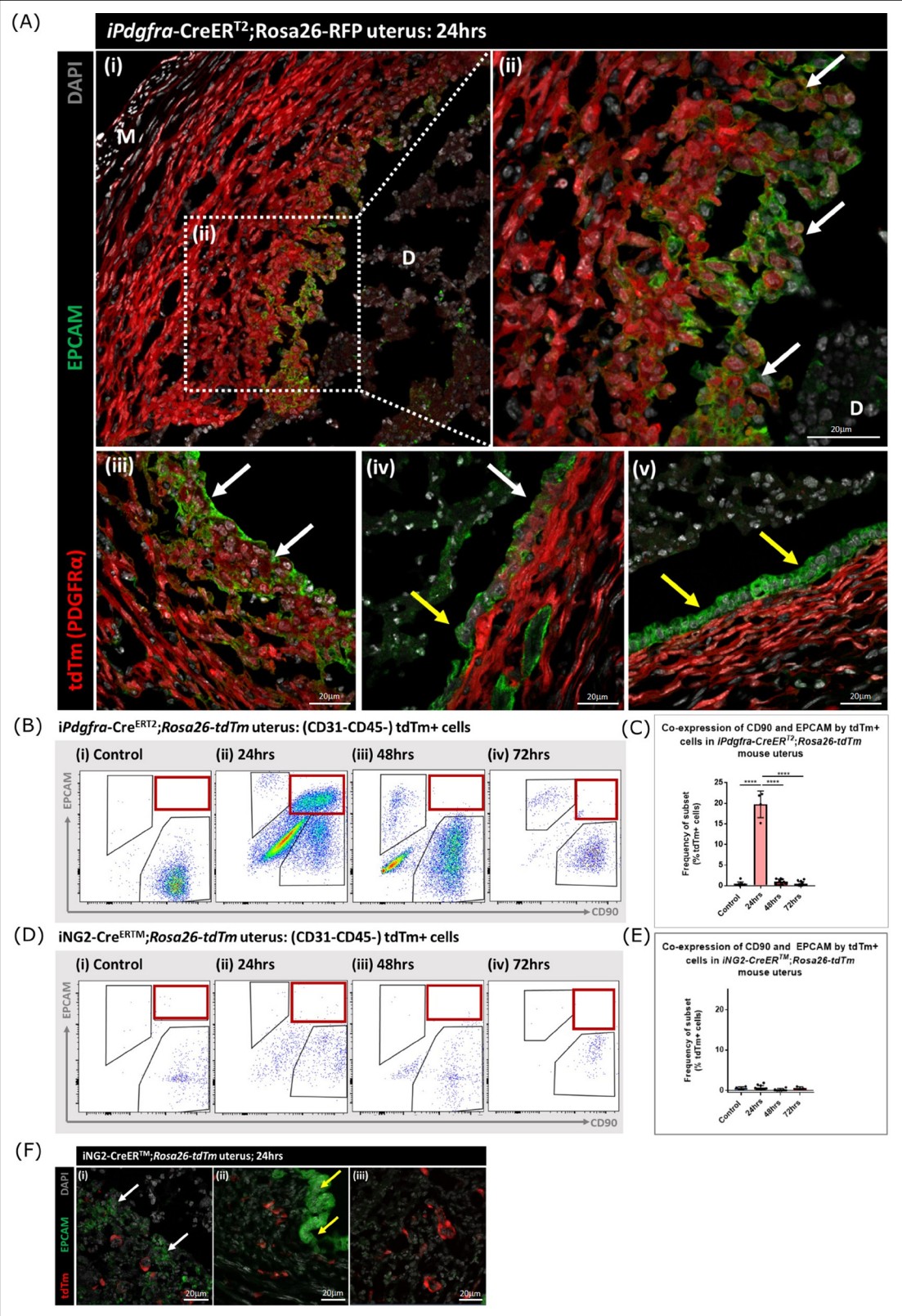

**Figure 6.** Lineage tracing of PDGFRα + cells identifies a population that undergoes MET. (**A**) Analysis of tdTm reporter protein and canonical epithelial cell marker EPCAM in *Pdgfra*-creERT2;*Rosa26*-tdTm uterine tissues 24 hr following progesterone withdrawal when the tissue is undergoing simultaneous decidual breakdown and repair. (i-iii) tdTm + EPCAM + cells can be detected in regions where the decidualised tissue is detached and the underlying stromal surface left exposed (white arrows) and (iv) in regions where a new epithelial layer is formed (white arrows) adjacent to tdTm-

*Figure 6 continued on next page*

*Figure 6 continued*

epithelial cells (yellow arrows). (**v**) Regions of epithelium where no tdTm + EPCAM + cells were detected exist in other regions of the tissue (yellow arrows). (**B**) FC: analysis of mesenchymal marker CD90 and epithelial marker EPCAM by tdTm + cells in *Pdgfra*-creERT2;*Rosa26*-tdTm uterine tissues 24/48/72 hr following progesterone withdrawal, detection of tdTm + CD90+EPCAM + cells in 24 hr tissues, red box. (**C**) Bar plot: quantification of tdTM + CD90+EPCAM + cells at 24/48/72 hr in *Pdgfra*-creERT2;*Rosa26*-tdTm uterine tissue, calculated as a frequency of tdTm + cells: Control (n=9): 0.45 ± 0.51%; 24 hr (n=4): 19.76 ± 3.24%; 48 hr (n=9): 0.96 ± 0.64%; 72 hr (n=13): 0.48 ± 0.54%; one-way ANOVA; Tukey's multiple comparisons. (**D**) FC: analysis of mesenchymal marker CD90 and epithelial marker EPCAM by tdTm + cells in iNG2-Cre[ERTM];*Rosa26*-tdTm uterine tissues 24/48/72 hr following progesterone withdrawal, no detection of tdTm + CD90+EPCAM + cells in 24 hr tissues, red box. (**E**) Bar plot: quantification of tdTm + CD90+EPCAM + cells in 24/48/72 hr iNG2-Cre[ERTM];*Rosa26*-tdTm uterine tissues,, calculated as a frequency of tdTm + cells: Control (n=4): 0.58 ± 0.34%; 24 hr (n=12): 0.64 ± 0.55%; 48 hr (n=6): 0.18 ± 0.29%; 72 hr (n=3): 0.61 ± 0.32%; one-way ANOVA; Tukey's multiple comparisons. (**F**) IHC: analysis of tdTm reporter protein and canonical epithelial cell marker EPCAM in iNG2-Cre[ERTM];*Rosa26*-tdTm uterine tissues 24 hr following progesterone withdrawal when the tissue is undergoing simultaneous decidual breakdown and repair. tdTm + EPCAM and tdTm-EPCAM + cells can be detected in (**i**) regions where the decidualised tissue is detached and the underlying stromal surface left exposed (white and yellow arrows respectively), (ii) regions where a new epithelial layer is formed and (iii) regions of residual epithelium. No co-localisation of tdTm and EPCAM is detected in any region throughout the tissue.

The online version of this article includes the following source data for figure 6:

**Source data 1.** Summary statistics for flow cytometry analyses performed in *Figure 6C*.

**Source data 2.** One-way ANOVA with Tukey's multiple comparisons test of values in *Figure 6C*.

**Source data 3.** Summary statistics for flow cytometry analyses performed in *Figure 6E*.

**Source data 4.** One-way ANOVA with Tukey's multiple comparisons test of values in *Figure 6E*.

Importantly we never detected, an equivalent population of tdTm + EPCAM + cells in NG2-creER/*Rosa26*-tdTm mice at any timepoint (*Figure 7—figure supplement 1A–E*).

## Discussion

In this study, we used a combination of single cell RNA sequencing and cell lineage tracing to investigate the contribution of the stromal mesenchyme to rapid restoration of the endometrial luminal epithelium at the time of menstruation. The human endometrium is unusual in its ability to experience cyclical episodes of breakdown/shedding (wounding) which occur in parallel with repair processes that leave the tissue intact without an obvious scar. We used a previously validated mouse model that mimics the key features of human menstruation in which we made some preliminary observations that aligned with studies suggesting MET of stromal cells might contribute to the rapid repair process (*Cousins et al., 2014*). In the work presented here we have identified a new population of repair-specific PDGFRb + cells that responded to changes in the endometrial environment induced by progesterone withdrawal by upregulating expression of genes associated with MET/epithelial cell identify. To best understand how this population might influence our understanding of endometrial repair we designed further experiments to answer two specific questions. First – what was the originating cell type of this transient population? Second – what was its fate?

The primary objective of the study was to explore whether we could substantiate our hypothesis that MET was occurring in stromal cells exposed by shedding of the decidual mass in our mouse model of menstruation (*Cousins et al., 2014*). Therefore we decided to use *Pdgfrb*-BAC-eGFP reporter mice in our menstrual model as we had previously shown eGFP was only expressed in stromal cell populations during the oestrus cycle (*Kirkwood et al., 2021*). Comparison between scRNAseq datasets from cycling mice identified three populations of mesenchyme cells that were unique to endometrial tissue recovered 24 or 48 hr after withdrawal of progesterone at a time when the tissue in the process of active breakdown/repair. One population was very closely associated with the previously identified pericyte cell population with sequence analysis suggesting it might represent a subgroup of cells that had some minor changes in function possibly associated with loss of some of the vascular compartment. A second population was only present at 48 hr and based on expression of several genes associated with apoptosis we concluded that it represented stromal cells that were not going to survive in repairing tissue and that cell death was one potential fate of the GFP + cells. The most interesting population of GFP + cells (F4) were unique to the 24 hr tissue with a transcriptome that included expression of both canonical mesenchymal and epithelial cell markers as well as a number of genes such as *Ctnnb1* (beta-catenin) and *Twist* which have been implicated cell transitions between mesenchyme and epithelium. Notably it has previously been reported that mice with stabilisation of

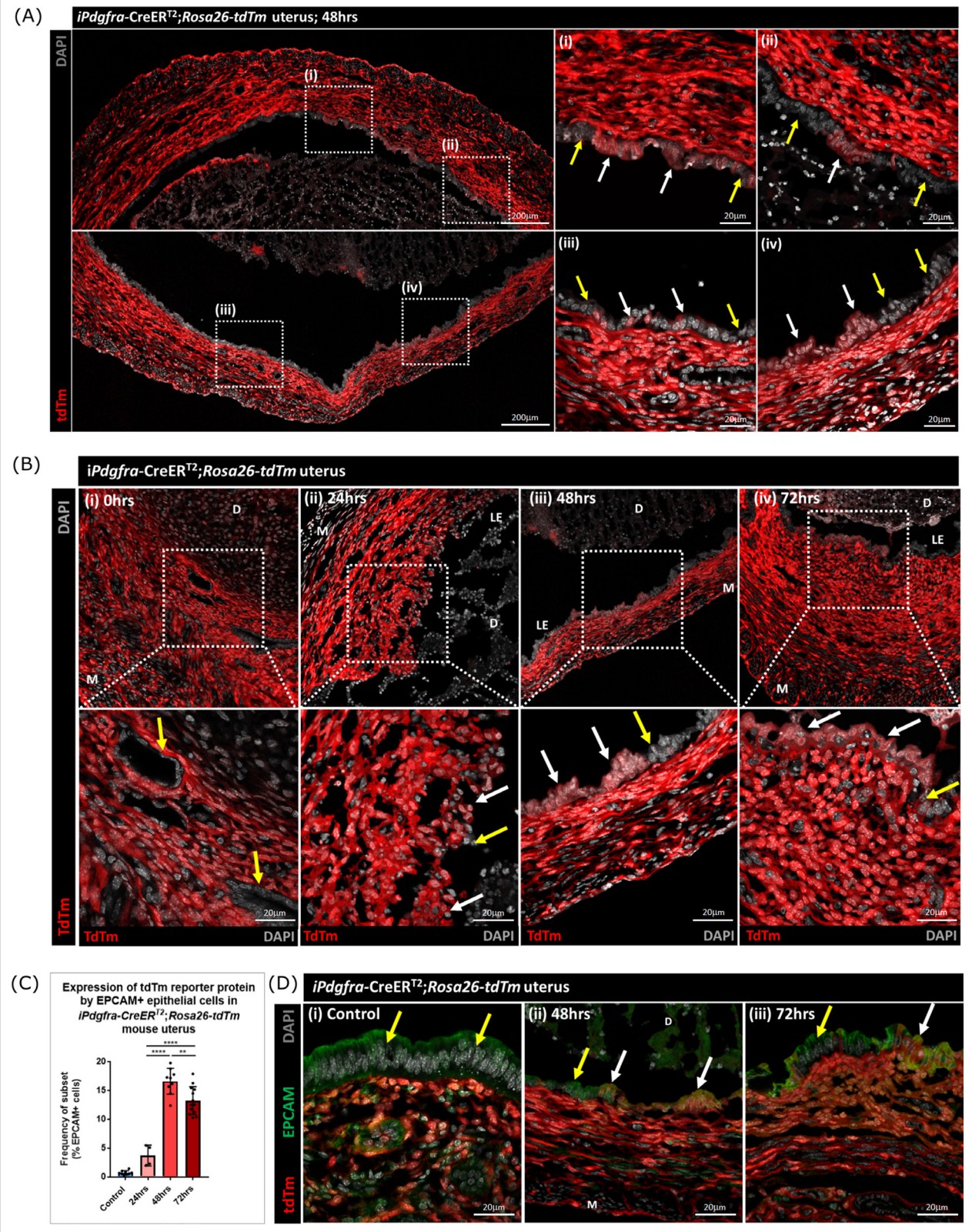

**Figure 7.** Lineage tracing studies confirm that epithelial cells derived from PDGFRα+fibroblasts persist in the post repair luminal epithelium. (**A**) Analysis of tdTm reporter protein expression in *Pdgfra*-creERT2;*Rosa26*-tdTm uterine tissues 48 hr following progesterone withdrawal when the tissue is remodelling and the luminal epithelium is re-instated. (**i–iv**) tdTm is expressed in stromal fibroblasts throughout the stromal tissue and in some cells in the luminal epithelium (white arrows) adjacent to tdTm- epithelial cells (yellow arrows). (**B**) IHC: analysis of tdTm reporter protein expression in *Pdgfra*-

*Figure 7 continued on next page*

*Figure 7 continued*

creERT2;*Rosa26*-tdTm uterine tissues 0, 24, 48, and 72 hr following progesterone withdrawal, tdTm + cells are detected within the renewed luminal epithelium at both 48 and 72 hr (white arrows) indicative of long term survival in the new epithelial layer. (**C**) Bar plot: quantification of FC data analysing EPCAM + cells that express tdTm in *Pdgfra*-creERT2;*Rosa26*-tdTm uterine tissues 24, 48, and 72 hr following progesterone withdrawal, calculated as a frequency of EPCAM + cells: Control (n=9): 0.75 ± 0.35%; 24 hr (n=4): 3.73 ± 1.81%; 48 h (n=9): 16.63 ± 2.24%; 72 hr (n=13): 13.26 ± 2.42%; one-way ANOVA; Tukey's multiple comparisons. (**D**) IHC: analysis of tdTm reporter and epithelial marker EPCAM expression in *Pdgfra*-creERT2;*Rosa26*-tdTm uterine tissues 48 and 72 hr following progesterone withdrawal. (ii-iii) tdTm + EPCAM + cells can be detected in the luminal epithelium at 48/72 hr (white arrows) adjacent to tdTm-EPCAM + epithelial cells (yellow arrows) and are not detected in (i) steady state control tissues.

The online version of this article includes the following source data and figure supplement(s) for figure 7:

**Source data 1.** Summary statistics for flow cytometry analyses performed in *Figure 7E*.

**Source data 2.** One-way ANOVA with Tukey's multiple comparisons test for values in *Figure 7C*.

**Figure supplement 1.** Evaluation of cell fate in NG2creER/Rosa26-tdTM reporter mice.

**Figure supplement 1—source data 1.** Sidak's multiple comparisons test of data from expression of tdTm reporter protein by EPCAM +epithelial cells in mouse uterus following induction with PDGFRalpha or NG2 Cre (shown in D).

*Ctnnb1*, a gene that plays an important role in Wnt signalling, had smaller uteri, endometrial gland defects, and were infertile (*Stewart et al., 2013*). Further analysis using flow cytometry and immuno-histochemistry confirmed the existence of GFP + EPCAM + cells in the 24 hr samples.

These results backed up our previous observation that MET occurred in regions of tissue associated with endometrial repair and are in agreement with studies exploring the fate of mesenchyme cells after parturition (*Huang et al., 2012*; *Patterson et al., 2013*). For example, Huang et al used a knock-in strategy with cells expressing LacZ induced by *Amhr2-Cre*: in virgin females the reporter was specific to stromal cells but post-partum they found labelled cells in both luminal and glandular epithelium. In another study using a SM22a-Cre driver, expression of which was reported to be restricted to a subset of CD34 +mesenchymal cells, reporter gene expression (GFP) was also detected in the epithelium (*Yin et al., 2019*). Whilst authors of all these papers concluded their results were consistent with stromal MET this interpretation has been challenged (*Ghosh et al., 2020*). Specifically, Ghosh et al conducted a comprehensive examination of embryonic and adult reproductive tracts using LacZ reporter lines driven by promoters for the genes *Amhr2, Sm22, Cspg4, Thy1* and *Pdgfrß* all of which are expressed in mesenchyme. They specifically questioned whether epithelial cells expressing reporter proteins in the adult endometrium arose from MET or were induced at a time when cells had meso-epithelial characteristics. In all cases, they attributed epithelial expression in adult epithelial cells to activation of the promoters during embryonic life ruling out MET in adult cycling mice (*Ghosh et al., 2020*).

One limitation of the results obtained using *Pdgfrb*-BAC-eGFP reporter mice was that the knockin strategy used to generate the line meant GFP was only expressed in cells when the *Pgdfrb* promoter was active. We have previously established that the expression of GFP in the endometrium of adult female mice was identical to the native PDGFRβ protein and was stroma-specific (*Kirkwood et al., 2021*). A comparison between GFP expression in cycling mice those using uteri recovered after induction of endometrial shedding/repair led us to conclude that the presence of GFP in the epithelial cells was most likely because the turnover of the GFP was slow enough for it to persist after the *Pgdfrb* promoter was no longer active. This would be consistent with reports that eGFP protein was engineered to have increased fluorescence and slower turnover in vivo *Zhao et al., 1999*; although the half-life of the protein varies in different cell models a paper exploring in vivo expression in cancer reported it to be ~15 hr (*Danhier et al., 2015*). These results were also considered consistent with detection of less GFP in EPCAM + epithelial cells at 48 than 24 hr when this was evaluated using flow cytometry.

Therefore, to make a more robust case for MET of endometrial stromal cells we expanded our scRNAseq datasets by analysing the epithelial cells (EPCAM+) from tissue recovered from both cycling mice and those in which endometrial repair was well advanced (48 hr). We then complemented these findings by undertaking lineage tracing studies in two new lines of mice using a strategy that induced expression of a tdTm reporter protein in adulthood to avoid any confounding effects of MET that might occur during development of the uterus. Analysis of EPCAM + cells resulted in identification of nine subpopulations of epithelial cells based on cluster analysis and comparisons to genes previously identified as being expressed in different locations including the lumen (*Yang et al., 2021*) and glands (*Foxa2, Kelleher et al., 2017*).

When we combined our eGFP + and EPCAM + scRNAseq datasets three broad groups of cell types was present: perivascular, fibroblast and epithelial with the repair-specific transient population of cells (24 hr specific, F4) appearing as a distinct fibroblast subpopulation which on trajectory analysis appeared to form a 'bridge' between the previously identified fibroblast clusters (*Kirkwood et al., 2021*) and the new epithelial datasets. Based on previous reports that identified cells with stem-like properties in a perivascular location (*Gargett et al., 2016*) we had initially anticipated pericytes might have been mobilised to undergo MET however the trajectory data suggested we might be wrong. To complement and extend the transcriptional profiling and to gain a fuller picture of the source of the putative MET cells we compared results from two new lines of mice. One in which we targeted fibroblasts using a Cre under the control of *Pdgfra* and a second line in which all perivascular/vSMC were targeted using the *Cspg4* (NG2) promoter. We used a TAM-dependent induction schedule that we refined so that there was robust transgene induction but no residual impact on the response of the tissue to decidualisation and induction of endometrial injury. The results obtained were unequivocal and supported not only MET of fibroblasts but also their incorporation into the restored luminal epithelium in a layer adjacent to epithelial cells which had not arisen by MET but were likely to have migrated from residual epithelial cell remnants left at time of shedding. These novel findings provide robust evidence that the endometrium can use multiple mechanisms including recruitment of mesenchyme cells to rapidly restore its integrity a feature of the tissue that is important for its normal functioning and for fertility. In future studies, it would be interesting to use cell ablation strategies specifically targeting F2 to investigate whether repair is delayed if they are not present, whether any other fibroblasts replace them and the relative contributions of the other mechanisms previously identified (epithelial proliferation and migration *Cousins et al., 2014*) to the acute phase of restoration of luminal epithelial integrity.

The generation of these scRNAseq datasets have allowed us to make some preliminary comparisons to results from studies using this technique to explore cell heterogeneity in human endometrium and human endometrial cells (*Wang et al., 2020*; *Queckbörner et al., 2021*; *Cao et al., 2021*; *Garcia-Alonso et al., 2021*; *Shih et al., 2022*; *Wu et al., 2022*). Queckborner and colleagues focused their analysis cells from 3 donors obtained during in the proliferative phase of their menstrual cycle (6864 cells) using a protocol that enriched for stromal cells (*Queckbörner et al., 2021*). They subtyped two populations of perivascular/pericyte cells with different expression levels of MY11D and CSPG4 both of which we have previously mapped to perivascular populations in the mouse endometrium (*Kirkwood et al., 2021*). They subtyped the stromal fibroblasts into 10 closely related clusters of fibroblasts and concluded that the markers for progenitor cells were not specific enough. They didn't analyse any tissues/cells from the menstrual phase so there are no comparable data to ours. In contrast, *Wang et al., 2020* incorporated tissues from across the menstrual cycle in their analysis including 6 that were assigned to the menstrual phase according to their supplementary data although these were identified as days 4–11 of the cycle which might be after the most rapid phase of repair (anticipated to be on days 1–3). They reported a previously uncharacterised ciliated epithelial subtype and changes in luminal epithelium and stroma that were discrete to the window of implantation but nothing specific to repair mechanisms. Cao et al focused their attention on the perivascular cell population performing sequence analysis of CD140b+CD146+ (eMSC) cells isolated from menstrual phase (days 2–3) and secretory phases (n=3 women in each group *Cao et al., 2021*). They identified two subclusters in cultured cells from the menstrual phase with the larger cluster having high expression of PDGFRβ/PDGFRα consistent with a fibroblast identity. When they compared their data to the Wang dataset they noted significant changes in gene expression patterns and concluded that the cell transcriptomes had been influenced by cell culture which may compromise attempts to compare between in vitro and in vivo data.

In a recent study, *Shih et al., 2022* used a single-cell approach to interrogate tissue recovered from menstrual effluent from patients with an without a diagnosis of endometriosis 9 of which were classified as 'controls' (8/9 had normal cycle length 26-31d). Cluster analysis showed a high proportion of immune cells, five closely related stromal cell populations and three epithelial cell clusters (*Shih et al., 2022*). Comparison between the genes that defined each subpopulation of stromal cells (GFBP1, MGP, IL11, SRGN, HSPA6) with our current dataset did not find any that corresponded to F4 which is probably unsurprising as these signatures were based on shed cells not those that remained in situ and therefore able to contribute to the new luminal epithelium. In contrast, we did find some homologies

within the dataset published by *Wu et al., 2022* whose analysis included single-cell sequencing of full thickness endometrium from the proliferative and secretory phases (no menstrual tissue). Their stromal cell subclusters included one enriched for SFRP4 which they also showed provided a positive stimulus to regeneration in a rat endometrial injury model. Comparison with our own datasets identified *Sfrp4* expression was highest in our F2 and F4 populations, very low in F1/F3 and absent from the epithelial cells: two of the factors secreted by the SFRP4 cells were also higher in the mouse F4 than other cell clusters (PENK, GLIPR1). Whilst further work is needed to validate whether a population equivalent to F4 is present in the menstrual endometrium looking for SFRP4 cells might provide a useful starting point and it is also notable that the authors found the SFRP4 cells to be distinct to putative progenitors identified as SUSD2 + which were enriched in the perivascular population.

In summary, we describe novel data and new datasets that show MET can contribute to the rapid scar-free healing of the endometrial luminal epithelium when this occurs in response to an endometrial 'wound' at the time of menstruation. The cell type involved in this MET is a PDGFRα+stromal cell and not the NG2 + pericytes. Further studies on the origin of these cells and their contribution to repair will require new studies using CRE lines which are specific to F2 for lineage tracing and cell ablation. The current findings also provide a platform for further analysis of the fibrosis-resistant phenotype of endometrial fibroblasts, comparisons to conditions where their function is abnormal (Asherman's syndrome, endometriosis) and comparison to other mucosal barrier tissues such as that found in the oral cavity as recently described by *Williams et al., 2021*. We postulate that manipulation of putative MET progenitors/specific cell types in the disease setting may provide novel strategies in the management and/or treatment of disorders associated with poor response to healing or excess fibrosis.

# Materials and methods
## Mouse lines
In *Pdgfrb*-BAC-eGFP reporter mice, eGFP expression is driven by the regulatory sequences of the *Pdgfrb* gene and is therefore expressed by all cells in which the promoter is active. Details of these transgenic mice and their use for studies in the cycling mouse endometrium have been previously described (*Kirkwood et al., 2021*). These mice were originally obtained from GENSAT and deposited in MMRRC-STOCK Tg(*Pdgfrb*-EGFP)JN169Gsat/Mmucd, 031796-UCD. Founder stocks were obtained from Professor Neil Henderson (CIR, University of Edinburgh).

For lineage-tracing experiments mice that express Cre recombinase in response to a Tamoxifen inducible promoter were cross bred with the Ai14 reporter mouse. Briefly, Ai14 is a Cre reporter allele designed with a loxP-flanked STOP cassette preventing the transcription of a CAG promoter-driven red fluorescent protein variant (tdTomato; tdTm)- all inserted into *the Gt(ROSA)26Sor* locus (*Rosa26*). Ai14 mice express robust tdTm fluorescence following cre-mediated recombination. These mice were originally obtained from the Jackson Laboratory [STOCK *Gt(ROSA)26Sor*tm14(CAG-tdTomato)Hze]. Founder stocks were obtained from Dr David Ferenbach and Dr Bryan Conway (University of Edinburgh).

We had previously identified expression of NG2(*Cspg4*) and Pdgfrα as specific to endometrial perivascular and fibroblast populations of mesenchymal cells respectively (*Kirkwood et al., 2021*). We therefore selected mice which would express Cre recombinase in response to induction of the promoters of these genes as appropriate for segregating cells representing these two subpopulations in our mouse model of endometrial repair. *Pdgfra*Cre-ERT2 mice express a tamoxifen-inducible Cre recombinase (CreERT2) under the control of the mouse *Pdgfra* promoter/enhancer. The generation of *Pdgfra*Cre-ERT2 mice has been described previously (*Chung et al., 2018*). Founder stocks were a gift to Professor Neil Henderson from Professor Brigid LM Hogan and Dr Christina E Barkauskas (Duke University, USA). NG2Cre-ER BAC transgenic mice express a tamoxifen-inducible Cre recombinase (CreER) under the control of the mouse NG2 (*Cspg4*) promoter/enhancer. The generation of NG2Cre-ER mice has been previously described (*Zhu et al., 2011*). Founder stocks were obtained from the Jackson Laboratory [STOCK B6.Cg-Tg(Cspg4-cre/Esr1*)BAkik/J].

## Tamoxifen induction of tdTomato transgenes
Tamoxifen (Sigma, Cat. #T5648-1G, TMX) was dissolved in sesame seed oil to a concentration of 20 mg/ml and left overnight on a roller at 37 °C (light protected). For TMX administration, in initial screening experiments an intraperitoneal (IP) injection of 100 µl was given to each mouse (standard

20 g weight) for 3 consecutive days (TMX dose 100 mg/kg). In later experiments the method was refined and TMX was administered via oral gavage (daily, 3x100 µl/day). To optimise an appropriate TMX wash out period prior to the mouse model of 'menstruation' (endometrial shedding and repair), uterine weights and the stromal to epithelial cell ratio (as determined by flow cytometry analysis) were analysed in mice treated with TMX or vehicle (sesame oil) and uterine tissues collected 24 hr, 1 week, 4 weeks and 8 weeks later. After 4 weeks following the final tamoxifen administration, tissue analysis was comparable to controls (Vehicle treated, no TMX). To validate timing so that the mouse model of 'menstruation' still worked, the decidualisation response (≥1 uterine horn decidualised) and corresponding uterine weights were analysed in mice in which reporter expression was induced with TMX and uterine tissues collected 24, 48, and 72 hr following the withdrawal of progesterone (onset of 'menses) and compared to those where no TMX administration had occurred. No significant differences were observed indicating that a tamoxifen induction 4 weeks prior to the start of the mouse model of 'menstruation' had no deleterious effects on the model.

### Mouse model of 'menstruation' (endometrial shedding and repair)

All animal procedures were carried out in accordance with the Animal Welfare and Ethical Review Body (AWERB) and under licensed approval from the UK Home Office. The mouse model of 'menstruation' (*Figure 1A* (i)) was conducted according to standard protocols that have been described in detail in previous papers (*Cousins et al., 2016a*, *Cousins et al., 2016b*, *Cousins et al., 2014*). Briefly, mice between 8 and 12 weeks of age were ovariectomised (d0) and given daily injections of β-oestradiol (E2) in sesame oil (100 ng/100 µl, d7-9). A progesterone (P)-secreting pellet was inserted (subcutaneous) on d13; mice also received daily injections of E2 (5 ng/100 µl; d13-15). A critical step in the model is the transformation of stromal fibroblasts into cells with a decidual phenotype achieved by introduction of a small amount of oil via the vagina (d15). Removal of a progesterone-secreting pellet 4 days later (d19) results in a rapid fall in progesterone culminating in breakdown and shedding of the decidual mass leaving parts of the endometrial lining with a surface denuded of epithelial cells (24 hr; *Cousins et al., 2014*). Within 3 days (72 hr after pellet removal) the integrity of the luminal epithelial layer is fully restored and gross uterine architecture resembles that of mice prior to ovariectomy (*Figure 1A* (ii)).

In those studies using mice in which transgene expression was induced using TMX the protocol was modified to ensure there were no agonist or antagonist impacts of the drug on the endometrium (*Gielen et al., 2008*) which could alter responsiveness to exogenous E2. Pilot studies were conducted and a 4 w wash out period applied to all TMX-treated mice before they were used in the menstruation model.

The minimum number of experimental animals to provide statistically significant data has been used. Extensive experience with the mouse model of simulated menstruation protocol has found that an n=9 mice per experimental group is sufficient to detect meaningful changes in parameters of interest (cell frequency, cell number, gene expression). This is supported by data in published studies (*Cousins et al., 2014*; *Cousins et al., 2016a*) and in agreement with previous consultation on experimental design provided by an expert statistician to support past and current research grants. To mitigate against losses and the potential need to use non-parametric methods a group size of 10 mice per experimental group was used in experimental planning.

In the current study, while individual uterine horns were analysed as distinct samples given the potential for each to exhibit varying degrees of decidualisation in response to the stimulus, each mouse used in the protocol was considered an individual biological replicate to account for biological variation. To generate technical replicates, each protocol was performed at least twice per outcome being analysed ie time point of the post-menstrual tissue repair window being analysed.

### Tissue recovery and processing

To preserve and detect fluorescent signal from transgenes, uteri were fixed in 4% (w/v) PFA for 2 hr at 4 °C, rinsed thoroughly in PBS and stored overnight in 18% (w/v) sucrose at 4 °C before embedding in OCT medium and storage (–80 °C). Frozen tissue sections (5 µm) were cut and mounted onto Xtra adhesive pre-cleaned micro slides (Surgipath, Leica Biosystems) and air-dried at room temperature for 1 hr prior to staining.

## Analysis of tissue sections for expression of fluorescent transgenes and proteins associated with specific lineages

Analysis of transgene expression was complemented by co-staining with selected antibodies according to established protocols (*Kirkwood et al., 2021*). Briefly, for immunofluorescence, tissue sections were washed in PBS to remove residual OCT and incubated with 3% (v/v) hydrogen peroxide solution in methanol for 30 min at room temperature, washed (1x5 min wash in PBS containing 0.05% (v/v) Tween20 and 2x5 min washes in PBS) and further incubated with 20% (v/v) normal goat serum for 30 min at room temperature. Sections were washed and incubated overnight at 4 °C with primary antibody at an optimised dilution in NGS; antibodies against EPCAM (epithelial cells), CD146 (endo-thelial/perivascular cells), PDGFRα (fibroblasts) and NG2 (perivascular cells) were used in this study (*Supplementary file 1a*). Following a further wash, sections were incubated with an HRP-conjugated secondary antibody at an optimised dilution in NGS for 1 hr at room temperature followed by a 10 min incubation with Opal Polaris 480 solution (Ayoka Biosciences). After staining, sections were counterstained with DAPI (Sigma, D9542), overlaid with Permaflour (Immunotech) and mounted with coverslips (VWR Prolabo) before imaging on a Zeiss LSM 510 Meta Confocal microscope (Zeiss).

## Cell isolation and purification using FACS

Tissue processing for flow cytometry analysis and FACS was performed as previously described (*Cousins et al., 2016a*, *Kirkwood et al., 2021*). Briefly, whole uterine horns were dissociated and incubated with collagenase (10 mg/ml) and DNase (10 mg/ml) for 30 min at 37 °C. Tissues were further dispersed and washed in FACS buffer (PBS Ca2-Mg2-: 5% charcoal stripped foetal calf serum (CSFCS), 2 mM EDTA) and subsequently strained through 70 μm and 40 μm cell strainers. Cell suspensions were centrifuged at 400rcf for 5 min and cell pellets re-suspended in 1 ml ACK lysing buffer (Gibco, Cat. No. A10492-01) for 1 min. Suspensions were then washed (as above) and incubated for 30 min on ice with optimised dilutions of fluorescently-conjugated antibodies as detailed in *Supplementary file 1b*. Cell suspensions were washed and re-suspended in PBS at 4 °C and analysed using a BD 5 L LSR Fortessa and BD FACSDiva software (BD Biosciences). To exclude dead cells, DAPI was added prior to flow cytometry analysis. Cells were sorted using a FACS Aria II instrument and BD FACSDiva software (BD Biosciences). Data analysis was performed using FlowJo analysis software (FlowJo LLC, Oregon, USA) and statistics carried out using GraphPad Prism. Samples were allocated to experimental groups based on the time that tissues were collected following withdrawal of progesterone in the mouse model of simulated menstruation protocol. For flow cytometry analyses samples were analysed without access to experimental group information to reduce bias in the data.

## Single-cell RNA sequencing- 10x genomics

Endometrial mesenchymal cells (GFP+) were isolated from *Pdgfrb*-BAC-eGFP mice 24 and 48 hr after removal of a progesterone pellet (times chosen to model active shedding/breakdown and active repair respectively; *Figure 1A*). For identification of definitive endometrial epithelial cell populations GFP-EPCAM + cells were isolated from *Pdgfrb*-BAC-eGFP mice 48 hr after removal of a progesterone pellet (time chosen to reflect complete re-epithelialisation) and cycling/control mice. Cells were recovered from 4 mice at each stage using FACS; 25,000 cells/sample giving a total of 100,000 cells for each population of interest for downstream application. Isolated cell suspensions were counted, and viability confirmed to be >85% using a TC20 Automated Cell Counter (BioRad, Cat. No. 1450102).

Cells were partitioned into nanolitre-scale Gel bead-in-Emulsions (GEMs) containing unique 10 x barcodes using the 10 x Chromium Controller (10 x Genomics, USA) and cDNA libraries generated as previously described in *Kirkwood et al., 2021*. Sequencing was carried out on the Illumina NovaSeq platform using bespoke 10 x parameters, according to the standard protocols at the facility (Edinburgh Genomics: http://genomics.ed.ac.uk/, Edinburgh).

Data files generated for mesenchymal cells (GFP+) from *Pdgfrb*-BAC-eGFP mice 24 and 48 hr after removal of a progesterone pellet were compared to cycling/control data files generated in previous studies which are available at GEO Series accession number GSE160772 (*Kirkwood et al., 2021*).

### Bioinformatic workflows

Raw sequencing data files were processed using 10 x Cell Ranger following standard protocols (version 2.0.1; https://www.10xgenomics.com). Genome alignment, filtering, barcode and UMI counting were

performed using the 'cellranger-count' function and 'refdata-cellranger-mm10-1.2.0' transcriptome as supplied by 10 x genomics. For downstream QC, clustering and gene expression analysis the *Seurat* R package (V3) (*Stuart and Satija, 2019*) was utilised with R version 4.0.2 in line with analyses performed as in *Kirkwood et al., 2021*. Briefly, the data was filtered based on standard QC metrics: number of unique genes detected in each cell, nFeature_RNA >200 & <5000; the percentage of reads that map to the mitochondrial genome (<10%); and the percentage of reads that map to ribosomal proteins (<10%). The *Scrublet* python package was used to further identify and remove putative doublets from scRNA seq data (*Wolock et al., 2019*). Following filtering, data was normalised and scaled and PCA analysis used to detect highly variable features/genes. Unsupervised clustering based on the first 20 principal components of the most variably expressed genes was performed using a graph based approach ('FindNeighbours', 'FindClusters'; resolution = 0.2). Clusters were visualised using the manifold approximation and projection (UMAP) method and identified by analysing the expression of canonical cell markers.

Differential gene expression analysis was performed ('FindAllMarkers') to identify genes expressed by each cell cluster when compared to all other clusters, using the non-parametric Wilcoxon rank sum test and p-value threshold of <0.05. Over-represented functional annotations in the differentially expressed genes were detected using the *clusterProfiler* package (*Yu et al., 2012*) using core functions to interpret data in the context of biological function, pathways and networks. To complement these analyses, annotated gene sets were downloaded from the Molecular Signatures Database (MSigDB) under the C5 ontology gene sets with a focus on genes associated with EMT/MET: 'GOBP_epithelial_to_mesenchymal_transition' and 'GOBP_mesenchymal_to_epithelial_transition' (*Subramanian et al., 2005*; *Liberzon et al., 2011*).

*Seurat* (V3) was used to integrate mesenchymal and epithelial scRNAseq datasets as described in *Stuart and Satija, 2019*. This method uses the FindIntegrationAnchors() function to identify cross-dataset 'anchors' ie. pairs of cells in each dataset that are in a matched biological state. These 'anchors' are then used to align the datasets using the IntegrateData() function. The standard *Seurat* integration workflow is used both to correct for technical differences between datasets (batch correction based on canonical correlation analysis (CCA) and influenced by mutual nearest neighbours (MNN)) and make comparisons across experimental conditions. Following integration, data processing, clustering and visualisation was performed as described above (normalisation, scaling, PCA analysis, clustering).

In silico trajectory analysis was performed using *Monocle3* (*Qiu et al., 2017*). Monocle3 works on the basis that cells in different functional 'states' express different sets of genes, and if/as cells move between these states, they undergo a process of transcriptional re-configuration where some genes are activated whilst others are silenced. Monocle3 uses an algorithm to learn if there is a sequence of gene expression changes between single cells that connect clusters through a putative differentiation trajectory. This analysis generates 'roots', 'branches' and 'end-points'; 'roots' are points in the trajectory/cluster where the transcriptome of individual cells is most similar to their neighbours that is the region in the cluster that may represent a differentiated phenotype; 'branches' represent cellular differentiation points that cells would go through to travel from one part of the trajectory to another; and 'end-points' are the final states of a predicted differentiation path representing the fully differentiated cells. RNA velocity analysis was performed using *scVelo* (*Bergen et al., 2020*) which uses a mathematical model of splicing kinetics to determine the directionality of genetic changes between neighbouring cells. Briefly, a ratio of pre-mature (unspliced) to mature (spliced) mRNA counts is calculated for each gene and velocities obtained as residuals from this ratio. The velocities are combined to estimate the future state of each cell and predict the directionality of cell-to-cell transitions.

The data discussed in this publication have been deposited in NCBI's Gene Expression Omnibus (*Edgar et al., 2002*) and are accessible through GEO series accession number GSE198556.

## Additional information

### Funding

| Funder | Grant reference number | Author |
| --- | --- | --- |
| Medical Research Council | MR/N013166/1 | Phoebe M Kirkwood |

| Funder | Grant reference number | Author |
|---|---|---|
| Medical Research Council | MR/N024524/1 | Phoebe M Kirkwood<br>Douglas A Gibson<br>Isaac Shaw |
| Wellcome Trust | 219542/Z/19/Z | Neil C Henderson |

The funders had no role in study design, data collection and interpretation, or the decision to submit the work for publication. For the purpose of Open Access, the authors have applied a CC BY public copyright license to any Author Accepted Manuscript version arising from this submission.

## Author contributions

Phoebe M Kirkwood, Conceptualization, Data curation, Formal analysis, Investigation, Visualization, Methodology, Writing - original draft, Writing – review and editing; Douglas A Gibson, Conceptualization, Supervision, Investigation, Writing – review and editing; Isaac Shaw, Data curation, Formal analysis, Investigation, Writing – review and editing; Ross Dobie, Conceptualization, Formal analysis, Investigation, Writing – review and editing; Olympia Kelepouri, Investigation; Neil C Henderson, Conceptualization, Supervision, Funding acquisition, Writing – review and editing; Philippa TK Saunders, Conceptualization, Resources, Data curation, Supervision, Funding acquisition, Writing - original draft, Project administration, Writing – review and editing

## Author ORCIDs

Philippa TK Saunders http://orcid.org/0000-0001-9051-9380

## Ethics

All animal experiments were performed under a license granted by the UK Home Office (PPL 70/8945) and were approved by the University of Edinburgh Animal Welfare and Ethical Review Body.

## Decision letter and Author response

Decision letter https://doi.org/10.7554/eLife.77663.sa1
Author response https://doi.org/10.7554/eLife.77663.sa2

# Additional files

## Supplementary files

• Supplementary file 1. Details of antibodies used in the study.
 (a) Primary and secondary antibodies with associated working dilutions used to detect proteins in mouse uterine tissues. (b) Flow cytometry antibodies selected and optimised to interrogate mesenchymal cell populations in murine uterus.

• MDAR checklist

## Data availability

Single cell RNAseq datasets have been deposited in GEO under accession codes GSE198556.

The following dataset was generated:

| Author(s) | Year | Dataset title | Dataset URL | Database and Identifier |
|---|---|---|---|---|
| Kirkwood PM, Gibson DA, Shaw I, Dobie R, Kelepouri O, Henderson NC, Saunders PT | 2022 | Single cell RNA sequencing and lineage tracing confirm mesenchyme to epithelial transformation (MET) contributes to repair of the endometrium at menstruation | https://www.ncbi.nlm.nih.gov/geo/query/acc.cgi?acc=GSE198556 | NCBI Gene Expression Omnibus, GSE198556 |

The following previously published dataset was used:

| Author(s) | Year | Dataset title | Dataset URL | Database and Identifier |
|---|---|---|---|---|
| Kirkwood PM, Gibson DA, Smith JR, Wilson-Kanamori JR, Kelepouri O, Zufiaurre AE, Dobie R, Henderson NC, Saunders PT | 2020 | Single cell RNA sequencing redefines the mesenchymal cell landscape of mouse endometrium [single cell RNA-seq] | https://www.ncbi.nlm.nih.gov/geo/query/acc.cgi?acc=GSE160772 | NCBI Gene Expression Omnibus, GSE160772 |

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
