## [Editor Report]

The investigators present an important finding showing the contribution of mesenchyme to epithelial transformation (MET) to the healing of the endometrial luminal epithelium, furthering our understanding of the mechanisms underlying endometrial regeneration, with a potential impact on related pathology.

---

## [Decision Letter]

**Decision letter after peer review:**

Thank you for submitting your article "Single cell RNA sequencing and lineage tracing confirm mesenchyme to epithelial transformation (MET) contributes to repair of the endometrium at menstruation" for consideration by *eLife*. Your article has been reviewed by 2 peer reviewers, including Shengtao Zhou as Reviewing Editor and Reviewer #2, and the evaluation has been overseen by Ricardo Azziz as the Senior Editor. The following individual involved in review of your submission has agreed to reveal their identity: Adam Stevens (Reviewer #1).

Essential revisions:

Reviewer 1 raises issues mainly around clarity. Suggestions for extra work are limited but would contribute extra clarity in the document if implemented. Reviewer 2 suggests checking whether this phenomenon could also be observed in human samples.

*Reviewer #1 (Recommendations for the authors):*

This work is an excellent example of integrating bioinformatic analysis and in vivo modelling. The synergy developed by this combined approach is apparent in the manuscript.

Comments:

I suggest that the authors may like to rethink some of the language used to describe similarity in the transcriptomic data. The comments on Figure 2A are a little too strong, for example – "suggesting this is not a phenotypically distinct cell population …" at the top of page 11. Statements concerning similarity at the level of differentially expressed genes based on the heat map can be fairly qualitative in their nature and should not be overstated.

Was there any other supporting evidence for cluster F5 being largely dead/apoptotic?

There are a couple of bioinformatic techniques that need a little more detail in the methods section. First is the use of GSEA supporting the data presented in Figure 2E. Second, and most importantly, is the approach used to combine datasets that result in the data presented in Figure 4 – this is very important to the work as it established the link between the fibroblast and epithelial clusters. Were batch effects considered in this analysis?

Figure 4D seems to imply a small number specific genes in establishing the epithelial aspect of similarity. Is further comment needed?

Monocle is not the only approach to trajectory analysis. Were others used as well? The authors may want to consider presenting RNA velocity analysis – I suspect it will support the monocle data.

Please clarify the meaning of "root" in describing the monocle analysis.

What do the grey dots mean in Figure 5A.

*Reviewer #2 (Recommendations for the authors):*

I have several key points to suggest the authors to consider prior to publication.

1. To further prove the clinical relevance of the findings, I would suggest the authors to check whether this phenomenon could also be observed in human samples.

2. As the menstrual period is intricately regulated by periodal hormonal change, is this subset of fibroblasts responsible of endometrial repair at menstruation also hormone-dependent?

3. Is it possible to identify the origin of this subset of fibroblasts? Whether they derive from differentiation of mesenchymal stem cells in the uterine wall or directly from differentiation of endometrial stem cells?

4. It would be recommended to further analyze the chemokine and cytokine profiles of this subset of fibroblasts to reveal their possible functions.

5. What would happen if this subset of fibroblasts are genetically depleted?

---

## [Author Response]

Reviewer #1 (Recommendations for the authors):This work is an excellent example of integrating bioinformatic analysis and in vivo modelling. The synergy developed by this combined approach is apparent in the manuscript.Comments:I suggest that the authors may like to rethink some of the language used to describe similarity in the transcriptomic data. The comments on Figure 2A are a little too strong, for example – "suggesting this is not a phenotypically distinct cell population …" at the top of page 11. Statements concerning similarity at the level of differentially expressed genes based on the heat map can be fairly qualitative in their nature and should not be overstated.

We agree we may have been a bit overenthusiastic in our comments and have made edits to the text as suggested which are highlighted on page 6.

Was there any other supporting evidence for cluster F5 being largely dead/apoptotic?

Evidence for cluster F5 being largely dead/apoptotic was purely based on *in silico* data. This included a high percentage of mitochondrial gene expression (% mito.genes) indicative of apoptotic/lysing cells, expression of key apoptosis-associated genes (*Cas3*, *Cas9*, *Trp53*, *Bak*) and GO analysis revealing DNA damage pathways. We have added text to the Results section to make it clear that this conclusion in the current paper is based on in silico data. We have also referenced our previous study using the same model (Armstrong et al. 2017) which detected high levels of apoptosis (based on cleaved caspase staining) in the stromal compartment 24h after removal of the P4 pellet (page 6).

There are a couple of bioinformatic techniques that need a little more detail in the methods section. First is the use of GSEA supporting the data presented in Figure 2E.

References and details of the MSigDB datasets that were downloaded to contribute to the analysis reported in Figure 2E have been added to the methods section (page 17) to clarify that these were annotated datasets defined as being involved in EMT/MET.

Second, and most importantly, is the approach used to combine datasets that result in the data presented in Figure 4 – this is very important to the work as it established the link between the fibroblast and epithelial clusters. Were batch effects considered in this analysis?

Yes batch correction was part of the Seurat integration workflow and more details have been added to the M and M section on page 17.

Figure 4D seems to imply a small number specific genes in establishing the epithelial aspect of similarity. Is further comment needed?

In Figure 4D we explored the relationship between F4 and the other subclusters using a range of accepted canonical cell type-associated genes – the results obtained complemented the other bioinformatic data (e.g. Figure 4A) and showed F4 as expressing genes usually considered as associated with either mesenchymal or epithelial lineages. However, it was also noted that F4 did not express the whole array of keratins and mucins as the ‘pure’ epithelial populations do. We have added extra text to results on page 6 to clarify our interpretation of these findings.

Monocle is not the only approach to trajectory analysis. Were others used as well? The authors may want to consider presenting RNA velocity analysis – I suspect it will support the monocle data.

Yes we did other analyses but had not included them so we appreciate the suggestion. We have now added Velocity analysis to the paper as a new supplementary figure. Information about the results of the trajectory analysis have been added (page 8) plus methods (page 17). The velocity analysis backs up the monocle data by showing F4 originating in the F2 subpopulation.

Please clarify the meaning of "root" in describing the monocle analysis.

We apologise for being too brief in the methods. Monocle3 ‘learns’ if there is a sequence of gene expression changes between neighbouring cells and builds the trajectory based on this. Once it learns overall trajectory it places cells at its position in this trajectory and this generates roots and branches. Roots are points in the trajectory/cluster where the cells are most similar to each other in terms of transcriptome ie. the region in the cluster that might represent a differentiated/distinct phenotype. Branches represent cellular decisions/differentiation points that cells would go through to travel from one part of the trajectory to another. Extra text has been added to the methods so that the reader will better understand what the figure shows (page 17).

What do the grey dots mean in Figure 5A.

Black dots are the roots, black lines are the trajectory branches and grey dots are ends of the inferred trajectories ie. the end-point of a differentiation path representing a cluster of fully differentiated cells. In addition to information in methods we have updated the legend to figure 5 to explain the colours of dots/lines.

Reviewer #2 (Recommendations for the authors):I have several key points to suggest the authors to consider prior to publication.1. To further prove the clinical relevance of the findings, I would suggest the authors to check whether this phenomenon could also be observed in human samples.

We regularly evaluate all the human studies that are published on endometrium some of which we have mentioned in our discussion. One of the challenges is that many studies lack sufficient depth of read in scRNAseq to identify subpopulations of stromal cells and it is rare for them to include samples from late secretory/menstrual samples. However since our paper was submitted we have found 2 new papers with some scRNAseq data that may be relevant – one of these Shih et al. (BMC Med 2022) analysed cells in menstrual effluent and the other Wu et al. (Cell Discovery 2022) identified an SFRP4+ stromal cell subpopulation. The Wu et al. paper is particularly interesting as when we searched our dataset for this gene (as well as factors made by these cells) we found that they were expressed in both F2 and F4. We have included these papers and discussed their findings in an additional paragraph added to the Discussion (page 12/13).

2. As the menstrual period is intricately regulated by periodal hormonal change, is this subset of fibroblasts responsible of endometrial repair at menstruation also hormone-dependent?

Results from our study (and those of other investigators) have shown that luminal epithelial repair during the initial phase of menstruation can occur independent of E2 and at a time when progesterone is low – our mice are ovx and ovarian hormone levels are low in women at this time. We have not investigated the hormone dependence of F2 in intact mice during the normal cycle but based on our previous findings using human stromal fibroblasts we would anticipate responses to androgens, oestrogens and progestins.

3. Is it possible to identify the origin of this subset of fibroblasts? Whether they derive from differentiation of mesenchymal stem cells in the uterine wall or directly from differentiation of endometrial stem cells?

We designed our experiments to focus on the acute phase of experimental repair – first to see if we could identify any population(s) that might transform into epithelia (using PDGFRb-GFP as a global marker of mesenchyme which included putative progenitor cells identified in previous studies) and then to follow up on the initial findings by using lineage tracing for two different mesenchyme populations (fibroblasts vs pericytes/SMC). When we started the study we thought a progenitor population might be the one contributing to epithelial repair but we found no evidence of this (see Discussion). Our study does not address the origin of the fibroblasts in the F2 population as the lineage tracing only targeted cells induced 1 month previously but we speculate this could be addressed by performing longer term lineage tracing studies focused on the fate of progenitors/stem cells which may reside closer to the myometrium (basal compartment).

4. It would be recommended to further analyze the chemokine and cytokine profiles of this subset of fibroblasts to reveal their possible functions.

Our analysis strategy was informed by the results of the informatics and as shown in the GO analysis illustrated in Figure 2B suggested the main functional processes associated with the F4 transcriptome were response to wounding, differentiation, epithelium differentiation, response to hypoxia all of which led us to focus on more towards MET/EMT than inflammation. In our previous evaluation of mesenchyme subsets in the cycling endometrium (Kirkwood et al. 2021, FASEBJ) we found the pericytes (P1) and the F1 subpopulation of fibroblasts had transcription profiles which matched to GO terms associated with regulation of immune response so we did not do further work on this aspect in the current paper.

5. What would happen if this subset of fibroblasts are genetically depleted?

That is an interesting question that would need to be addressed in a future study. In the study in which we first identified MET as a possible mechanism contributing to repair (Cousins et al. 2014) we also highlighted roles for epithelial proliferation and epithelial migration in the rapid restoration of the lumen (in agreement with results in human studies).

Based on these findings we speculate deletion of F2 would most likely delay but not completely block the repair process. We have added some extra text to the discussion to follow up on your question (page 13).